# A Novel Analysis Framework of Lower Complexity Bounds for Finite-Sum Optimization

## Abstract

This paper studies the lower bound complexity for the optimization problem whose objective function is the average of $n$ individual smooth convex functions. We consider the algorithm which gets access to gradient and proximal oracle for each individual component. For the strongly-convex case, we prove such an algorithm can not reach an $\varepsilon$-suboptimal point in fewer than $\Omega((n + \sqrt{\kappa n}) \log(1/\varepsilon))$ iterations, where $\kappa$ is the condition number of the objective function. This lower bound is tighter than previous results and perfectly matches the upper bound of the existing proximal incremental first-order oracle algorithm Point-SAGA. We develop a novel construction to show the above result, which partitions the tridiagonal matrix of classical examples into $n$ groups to make the problem difficult enough to stochastic algorithms. This construction is friendly to the analysis of proximal oracle and also could be used in general convex and average smooth cases naturally.

## 1 Introduction

We consider the minimization of the following optimization problem

$$\min_{\boldsymbol{x}\in\mathbb{R}^d} f(\boldsymbol{x}) \triangleq \frac{1}{n} \sum_{i=1}^{n} f_i(\boldsymbol{x}), \tag{1}$$

where the $f_i(\boldsymbol{x})$ are $L$-smooth and $\mu$-strongly convex. Accordingly, the condition number is defined as $\kappa = L/\mu$, which is typically larger than $n$ in real-world applications. Many machine learning models can be formulated as the above problem such as ridge linear regression, ridge logistic regression, smoothed support vector machines, graphical models, etc. This paper focuses on the first order methods for solving Problem (1), which access to the Proximal Incremental First-order Oracle (PIFO) for each individual component, that is,

$$h_f(\boldsymbol{x}, i, \gamma) \triangleq \left[ f_i(\boldsymbol{x}), \nabla f_i(\boldsymbol{x}), \operatorname{prox}_{f_i}^{\gamma}(\boldsymbol{x}) \right], \tag{2}$$

where $i \in \{1, \ldots, n\}$, $\gamma > 0$, and the proximal operation is defined as

$$\operatorname{prox}_{f_i}^{\gamma}(\boldsymbol{x}) = \arg\min_{\boldsymbol{u}} \left\{ f_i(\boldsymbol{u}) + \frac{1}{2\gamma} \|\boldsymbol{x} - \boldsymbol{u}\|_2^2 \right\}.$$

We also define the Incremental First-order Oracle (IFO)

$$g_f(\boldsymbol{x}, i, \gamma) \triangleq [f_i(\boldsymbol{x}), \nabla f_i(\boldsymbol{x})].$$

PIFO provides more information than IFO and it would be potentially more powerful than IFO in first order optimization algorithms. Our goal is to find an $\varepsilon$-suboptimal solution $\hat{\boldsymbol{x}}$ such that

$$f(\hat{\boldsymbol{x}}) - \min_{\boldsymbol{x}\in\mathbb{R}^d} f(\boldsymbol{x}) \leq \varepsilon$$

by using PIFO or IFO.

There are several first-order stochastic algorithms to solve Problem (1). The key idea to leverage the structure of $f$ is variance reduction which is effective for ill-conditioned problems. For example, SVRG (Zhang et al., 2013; Johnson and Zhang, 2013; Xiao and Zhang, 2014) can

find an $\varepsilon$-suboptimal solution in $\mathcal{O}((n+\kappa)\log(1/\varepsilon))$ IFO calls, while the complexity of the classical Nesterov's acceleration (Nesterov, 1983) is $\mathcal{O}(n\sqrt{\kappa}\log(1/\varepsilon))$. Similar results[1] also hold for SAG (Schmidt et al., 2017) and SAGA (Defazio et al., 2014). In fact, there exists an accelerated stochastic gradient method with $\sqrt{\kappa}$ dependency. Defazio (2016) introduced a simple and practical accelerated method called Point SAGA, which reduces the iteration complexity to $\mathcal{O}((n+\sqrt{\kappa n})\log(1/\varepsilon))$. The advantage of Point SAGA is in that it has only one parameter to be tuned, but the iteration depends on PIFO rather than IFO. Allen-Zhu (2017) proposed the Katyusha momentum to accelerate variance reduction algorithms, which achieves the same iteration complexity as Point-SAGA but only requires IFO calls.

The lower bound complexities of IFO algorithms for convex optimization have been well studied (Agarwal and Bottou, 2015; Arjevani and Shamir, 2015; Woodworth and Srebro, 2016; Carmon et al., 2017; Lan and Zhou, 2017; Zhou and Gu, 2019). Specifically, Lan and Zhou (2017) showed that at least $\Omega((n+\sqrt{\kappa n})\log(1/\varepsilon))$ IFO calls[2] are needed to obtain an $\varepsilon$-suboptimal solution for some complicated objective functions. This lower bound is optimal because it matches the upper bound complexity of Katyusha (Allen-Zhu, 2017).

It would be interesting whether we can establish a more efficient PIFO algorithm than IFO one. Woodworth and Srebro (2016) provided a lower bound $\Omega(n+\sqrt{\kappa n}\log(1/\varepsilon))$ for PIFO algorithms, while the known upper bound of the PIFO algorithm Point SAGA [3] is $\mathcal{O}((n+\sqrt{\kappa n})\log(1/\varepsilon))$. The difference of dependency on $n$ implies that the existing theory of PIFO algorithm is not perfect. This gap can not be ignored because the number of components $n$ is typically very large in many machine learning problems. A natural question is can we design a PIFO algorithm whose upper bound complexity matches Woodworth and Srebro's lower bound, or can we improve the lower bound complexity of PIFO to match the upper bound of Point SAGA.

In this paper, we prove the lower bound complexity of PIFO algorithm is $\Omega((n+\sqrt{\kappa n})\log(1/\varepsilon))$ for smooth and strongly-convex $f_i$, which means the existing Point-SAGA (Defazio, 2016) has achieved optimal complexity and PIFO can not lead to a tighter upper bound than IFO. We provide a novel construction, showing the above result by decomposing the classical tridiagonal matrix (Nesterov, 2013) into $n$ groups. This technique is quite different from the previous lower bound complexity analysis (Agarwal and Bottou, 2015; Woodworth and Srebro, 2016; Lan and Zhou, 2017; Zhou and Gu, 2019). Moreover, it is very friendly to the analysis of proximal operation and easy to follow. We also use this technique to study general convex and average smooth cases (Allen-Zhu, 2018; Zhou and Gu, 2019), and extend our result to non-convex problems (see Appendix J).

## 2   OUR ANALYSIS FRAMEWORK

In this paper, we consider the Proximal Incremental First-order Oracle (PIFO) algorithm for smooth convex finite-sum optimization. All proofs in this section can be found in Appendices C and D for a detailed version. We analyze the lower bounds of the algorithms when the objective functions are respectively strongly convex, general convex, smooth and average smooth (Zhou and Gu, 2019).

**Definition 2.1.** *For any differentiable function* $f : \mathbb{R}^m \to \mathbb{R}$,

- *$f$ is convex, if for any $\boldsymbol{x}, \boldsymbol{y} \in \mathbb{R}^m$ it satisfies $f(\boldsymbol{y}) \geq f(\boldsymbol{x}) + \langle \nabla f(\boldsymbol{x}), \boldsymbol{y} - \boldsymbol{x} \rangle$.*

- *$f$ is $\mu$-strongly convex, if for any $\boldsymbol{x}, \boldsymbol{y} \in \mathbb{R}^m$ it satisfies*

  $$f(\boldsymbol{y}) \geq f(\boldsymbol{x}) + \langle \nabla f(\boldsymbol{x}), \boldsymbol{y} - \boldsymbol{x} \rangle + \frac{\mu}{2}\|\boldsymbol{x} - \boldsymbol{y}\|_2^2.$$

- *$f$ is $L$-smooth, if for any $\boldsymbol{x}, \boldsymbol{y} \in \mathbb{R}^m$ it satisfies $\|\nabla f(\boldsymbol{x}) - \nabla f(\boldsymbol{y})\|_2 \leq L\|\boldsymbol{x} - \boldsymbol{y}\|_2$.*

---

[1]SVRG, SAG and SAGA only need to introduce the proximal operation for composite objective, that is, $f_i(\boldsymbol{x}) = g_i(\boldsymbol{x}) + h(\boldsymbol{x})$, where $h$ may be non-smooth. Their iterations only depend on IFO when all the $f_i(x)$ are smooth. Hence, we regard these algorithms only require IFO calls in this paper.

[2]Lan and Zhou's construction requires $f$ to be $\mu$-strongly convex and every $f_i$ to be convex, while this paper studies the lower bound with stronger condition that is every $f_i$ is $\mu$-strongly convex. For the same lower bound complexity, the result with stronger assumptions on the objective functions is stronger.

| | Upper Bounds | Previous Lower Bounds | Our Lower Bounds |
|---|---|---|---|
| $f_i$ is $L$-smooth and $\mu$-strongly convex with $n = \mathcal{O}(\kappa)$ | $\mathcal{O}\left((n + \sqrt{\kappa n}) \log(\frac{1}{\varepsilon})\right)$ (Allen-Zhu, 2017), IFO (Defazio, 2016), PIFO | $\Omega\left(n + \sqrt{\kappa n} \log(\frac{1}{\varepsilon})\right)$ (Woodworth and Srebro, 2016) PIFO | $\Omega\left((n + \sqrt{\kappa n}) \log(\frac{1}{\varepsilon})\right)$ [Theorem 3.1] PIFO |
| $f_i$ is $L$-smooth and $\mu$-strongly convex with $\kappa = \mathcal{O}(n)$ | $\mathcal{O}\left(n + \left(\frac{n}{1 + (\log(\frac{n}{\kappa}))_+}\right) \log\left(\frac{1}{\varepsilon}\right)\right)$ (Hannah et al., 2018), IFO | $\Omega\left(n + \left(\frac{n}{1 + (\log(\frac{n}{\kappa}))_+}\right) \log\left(\frac{1}{\varepsilon}\right)\right)$ (Hannah et al., 2018) IFO | $\Omega\left(n + \left(\frac{n}{1 + (\log(\frac{n}{\kappa}))_+}\right) \log\left(\frac{1}{\varepsilon}\right)\right)$ [Theorem 3.1] PIFO |
| $f_i$ is $L$-smooth and convex | $\mathcal{O}\left(n \log(\frac{1}{\varepsilon}) + \sqrt{\frac{nL}{\varepsilon}}\right)$ (Allen-Zhu, 2017) IFO | $\Omega\left(n + \sqrt{\frac{nL}{\varepsilon}}\right)$ (Woodworth and Srebro, 2016) PIFO | $\Omega\left(n + \sqrt{\frac{nL}{\varepsilon}}\right)$ [Theorem 3.3] PIFO |
| $\{f_i\}_{i=1}^n$ is $L$-average smooth and $f$ is $\mu$-strongly convex | $\mathcal{O}\left(\left(n + n^{\frac{3}{4}}\sqrt{\kappa}\right) \log\left(\frac{1}{\varepsilon}\right)\right)$ (Allen-Zhu, 2018) IFO | $\Omega\left(n + n^{\frac{3}{4}}\sqrt{\kappa} \log\left(\frac{1}{\varepsilon}\right)\right)$ (Zhou and Gu, 2019) IFO | $\Omega\left(\left(n + n^{\frac{3}{4}}\sqrt{\kappa}\right) \log\left(\frac{1}{\varepsilon}\right)\right)$ [Theorem 3.5] PIFO |
| $\{f_i\}_{i=1}^n$ is $L$-average smooth and $f$ is convex | $\mathcal{O}\left(n + n^{\frac{3}{4}}\sqrt{\frac{L}{\varepsilon}}\right)$ (Allen-Zhu, 2018) IFO | $\Omega\left(n + n^{\frac{3}{4}}\sqrt{\frac{L}{\varepsilon}}\right)$ (Zhou and Gu, 2019) IFO | $\Omega\left(n + n^{\frac{3}{4}}\sqrt{\frac{L}{\varepsilon}}\right)$ [Theorem 3.7] PIFO |

Table 1: We compare our PIFO lower bounds with existing results of IFO or PIFO algorithms, where $\kappa = L/\mu$. Note that the call of PIFO could obtain more information than IFO. Hence, any PIFO lower bound also can be regarded as an IFO lower bound, not vice versa.

**Definition 2.2.** *We say differentiable functions $\{f_i\}_{i=1}^n$, $f_i : \mathbb{R}^m \to \mathbb{R}$, to be L-average smooth if for any $\boldsymbol{x}, \boldsymbol{y} \in \mathbb{R}^m$, they satisfy*

$$\frac{1}{n} \sum_{i=1}^n \|\nabla f_i(\boldsymbol{x}) - \nabla f_i(\boldsymbol{y})\|_2^2 \leq L^2 \|\boldsymbol{x} - \boldsymbol{y}\|_2^2. \tag{3}$$

**Remark 2.3.** *We point out that*

1. *if each $f_i$ is L-smooth, then $\{f_i\}_{i=1}^n$ are L-average smooth.*

2. *if $\{f_i\}_{i=1}^n$ are L-average smooth, then $f(\boldsymbol{x}) = \frac{1}{n} \sum_{i=1}^n f_i(\boldsymbol{x})$ is L-smooth.*

We present the formal definition for PIFO algorithm.

**Definition 2.4.** *Consider a stochastic optimization algorithm $\mathcal{A}$ to solve Problem (1). Let $\boldsymbol{x}_t$ be the point obtained at time-step $t$ and the algorithm starts with $\boldsymbol{x}_0$. The algorithm $\mathcal{A}$ is said to be a PIFO algorithm if for any $t \geq 0$, we have*

$$\boldsymbol{x}_t \in \text{span}\left\{\boldsymbol{x}_0, \ldots, \boldsymbol{x}_{t-1}, \nabla f_{i_1}(\boldsymbol{x}_0), \cdots, \nabla f_{i_t}(\boldsymbol{x}_{t-1}), \text{prox}_{f_{i_1}}^{\gamma_1}(\boldsymbol{x}_0), \cdots, \text{prox}_{f_{i_t}}^{\gamma_t}(\boldsymbol{x}_{t-1})\right\}, \tag{4}$$

*where $i_t$ is a random variable supported on $[n]$ and takes $\mathbb{P}(i_t = j) = p_j$ for each $t \geq 0$ and $1 \leq j \leq n$ where $\sum_{j=1}^n p_j = 1$.*

Without loss of generality, we assume $\boldsymbol{x}_0 = \boldsymbol{0}$ and $p_1 \leq p_2 \leq \cdots \leq p_n$ to simplify our analysis. Otherwise, we can take $\{\hat{f}_i(\boldsymbol{x}) = f_i(\boldsymbol{x} + \boldsymbol{x}_0)\}_{i=1}^n$ into consideration. On the other hand, suppose that $p_{s_1} \leq p_{s_2} \leq \cdots \leq p_{s_n}$ where $\{s_i\}_{i=1}^n$ is a permutation of $[n]$. Define $\{\tilde{f}_i\}_{i=1}^n$ such that $\tilde{f}_{s_i} = f_i$, then $\mathcal{A}$ takes component $\tilde{f}_{s_i}$ in probability $p_{s_i}$, i.e., $\mathcal{A}$ takes $f_i$ in probability $p_{s_i}$.

To demonstrate the construction of adversarial functions, we first introduce the following class of matrices:

$$\boldsymbol{B}(m,\omega) = \begin{bmatrix} & & & & -1 & 1 \\ & & & -1 & 1 & \\ & & \cdot^{\cdot} & \cdot^{\cdot} & & \\ -1 & 1 & & & & \\ \omega & & & & & \end{bmatrix} \in \mathbb{R}^{m \times m}.$$

Then we define

$$\boldsymbol{A}(m,\omega) \triangleq \boldsymbol{B}(m,\omega)^{\top}\boldsymbol{B}(m,\omega) = \begin{bmatrix} \omega^2+1 & -1 & & & \\ -1 & 2 & -1 & & \\ & \ddots & \ddots & & \\ & & -1 & 2 & -1 \\ & & & -1 & 1 \end{bmatrix}. \tag{5}$$

The matrix $\boldsymbol{A}(m,\omega)$ is widely-used in the analysis of lower bounds for convex optimization (Nesterov, 2013; Agarwal and Bottou, 2015; Lan and Zhou, 2017; Carmon et al., 2017; Zhou and Gu, 2019). We now present a decomposition of $\boldsymbol{A}(m,\omega)$ based on Eq. (5).

Denote the $l$-th row of the matrix $\boldsymbol{B}(m,\omega)$ by $\boldsymbol{b}_l(m,\omega)^{\top}$ and let

$$\mathcal{L}_i = \big\{l : 1 \le l \le m, l \equiv i - 1 (\mathrm{mod}\ n)\big\}, \quad i = 1, 2, \cdots, n.$$

Our construction is based on the following class of functions

$$r(\boldsymbol{x}; \lambda_0, \lambda_1, \lambda_2, m, \omega) \triangleq \frac{1}{n}\sum_{i=1}^{n} r_i(\boldsymbol{x}; \lambda_0, \lambda_1, \lambda_2, m, \omega),$$

where

$$r_i(\boldsymbol{x}; \lambda_0, \lambda_1, \lambda_2, m, \omega) = \begin{cases} \lambda_1 \sum\limits_{l \in \mathcal{L}_1} \left\|\boldsymbol{b}_l(m,\omega)^{\top}\boldsymbol{x}\right\|_2^2 + \lambda_2 \left\|\boldsymbol{x}\right\|_2^2 - \lambda_0 \langle \boldsymbol{e}_m, \boldsymbol{x} \rangle, & \text{for } i = 1, \\ \lambda_1 \sum\limits_{l \in \mathcal{L}_i} \left\|\boldsymbol{b}_l(m,\omega)^{\top}\boldsymbol{x}\right\|_2^2 + \lambda_2 \left\|\boldsymbol{x}\right\|_2^2, & \text{for } i = 2, 3, \cdots, n. \end{cases} \tag{6}$$

We can determine the smooth and strongly-convex coefficients of $r_i$ as follows.

**Proposition 2.5.** *For any $\lambda_1 > 0, \lambda_2 \ge 0, \omega < \sqrt{2}$, we have that the $r_i$ are $(4\lambda_1 + 2\lambda_2)$-smooth and $\lambda_2$-strongly convex, and $\{r_i\}_{i=1}^{n}$ is $L'$-average smooth where*

$$L' = 2\sqrt{\frac{4}{n}\left[(\lambda_1 + \lambda_2)^2 + \lambda_1^2\right] + \lambda_2^2}.$$

We define the subspaces $\{\mathcal{F}_k\}_{k=0}^{m}$ as

$$\mathcal{F}_k = \begin{cases} \mathrm{span}\{\boldsymbol{e}_m, \boldsymbol{e}_{m-1}, \cdots, \boldsymbol{e}_{m-k+1}\}, & \text{for } 1 \le k \le m, \\ \{\boldsymbol{0}\}, & \text{for } k = 0. \end{cases}$$

The following technical lemma plays a crucial role in our proof.

**Lemma 2.6.** *For any $\lambda_0 \ne 0, \lambda_1 > 0, \lambda_2 \ge 0$ and $\boldsymbol{x} \in \mathcal{F}_k$, $0 \le k < m$, we have that*

$$\nabla r_i(\boldsymbol{x}; \lambda_0, \lambda_1, \lambda_2, m, \omega) \text{ and } \mathrm{prox}_{r_i}^{\gamma}(\boldsymbol{x}) \in \begin{cases} \mathcal{F}_{k+1}, & \text{if } k \equiv i - 1 (\mathrm{mod}\ n), \\ \mathcal{F}_k, & \text{otherwise.} \end{cases}$$

In short, if $\boldsymbol{x} \in \mathcal{F}_k$ and let $f_i(\boldsymbol{x}) \triangleq r_i(\boldsymbol{x}; \lambda_0, \lambda_1, \lambda_2, \omega)$, then there exists only one $i \in \{1, \ldots, n\}$ such that $h_f(\boldsymbol{x}, i, \gamma)$ could (and only could) provide additional information in $\mathcal{F}_{k+1}$. The "only one" property is important to the lower bound analysis for first order stochastic optimization algorithms (Lan and Zhou, 2017; Zhou and Gu, 2019), but these prior constructions only work for IFO rather than PIFO.

Lemma 2.6 implies that $\boldsymbol{x}_t = \boldsymbol{0}$ will host until algorithm $\mathcal{A}$ draws the component $f_1$. Then, for any $t < T_1 = \min_t\{t : i_t = 1\}$, we have $\boldsymbol{x}_t \in \mathcal{F}_0$ and $\boldsymbol{x}_{T_1} \in \mathcal{F}_1$. The value of $T_1$ can be regarded as the smallest integer such that $\boldsymbol{x}_{T_1}$ could host. Similarly, we can define $T_k$ to be the smallest integer such that $\boldsymbol{x}_{T_k} \in \mathcal{F}_k$ could host. We give the formal definition of $T_k$ recursively and connect it to geometrically distributed random variables in the following corollary.

**Corollary 2.7.** *Let*

$$T_0 = 0, \quad and \quad T_k = \min_t \{t : t > T_{k-1}, i_t \equiv k \pmod{n}\} \ for \ k \geq 1. \tag{7}$$

*Then for any $k \geq 1$ and $t < T_k$, we have $\boldsymbol{x}_t \in \mathcal{F}_{k-1}$. Moreover, $T_k$ can be written as sum of $k$ independent random variables $\{Y_l\}_{1 \leq l \leq k}$, i.e., $T_k = \sum_{l=1}^{k} Y_l$, where $Y_l$ follows a geometric distribution with success probability $q_l = p_{l'}$ where $l' \equiv l \pmod{n}, 1 \leq l' \leq n$.*

The basic idea of our analysis is that we guarantee the minimizer of $r$ lies in $\mathcal{F}_m$ and assure the PIFO algorithm extend the space of $\mathrm{span}\{\boldsymbol{x}_0, \boldsymbol{x}_1, \ldots, \boldsymbol{x}_t\}$ slowly with $t$ increasing. We know that $\mathrm{span}\{\boldsymbol{x}_0, \boldsymbol{x}_1, \ldots, \boldsymbol{x}_{T_k}\} \subseteq \mathcal{F}_k$ by Corollary 2.7. Hence, $T_k$ is just the quantity that reflects how $\mathrm{span}\{\boldsymbol{x}_0, \boldsymbol{x}_1, \ldots, \boldsymbol{x}_t\}$ verifies. Because $T_k$ can be written as the sum of geometrically distributed random variables, we needs to introduce some properties of such random variables which derive the lower bounds of our construction.

**Lemma 2.8.** *Let $\{Y_i\}_{1 \leq i \leq N}$ be independent random variables, and $Y_i$ follows a geometric distribution with success probability $p_i$. Then*

$$\mathbb{P}\left(\sum_{i=1}^{N} Y_i > \frac{N^2}{4(\sum_{i=1}^{N} p_i)}\right) \geq 1 - \frac{16}{9N}. \tag{8}$$

From Lemma 2.8, the following result implies how many PIFO calls we need.

**Lemma 2.9.** *If $M \geq 1$ satisfies $\min_{\boldsymbol{x} \in \mathcal{F}_M} f(\boldsymbol{x}) - \min_{\boldsymbol{x} \in \mathbb{R}^m} f(\boldsymbol{x}) \geq 9\varepsilon$ and $N = n(M+1)/4$, then we have*

$$\min_{t \leq N} \mathbb{E}f(\boldsymbol{x}_t) - \min_{\boldsymbol{x} \in \mathbb{R}^m} f(\boldsymbol{x}) \geq \varepsilon.$$

## 3 MAIN RESULTS

We present the our lower bound results for PIFO algorithms and summarize all of results in Table 1 and 2 . We first start with smooth and strongly convex setting, then consider the general convex and average smooth cases.

**Theorem 3.1.** *For any PIFO algorithm $\mathcal{A}$ and any $L, \mu, n, \Delta, \varepsilon$ such that $\kappa = L/\mu \geq 2$, and $\varepsilon/\Delta \leq 0.5$, there exist a dimension $d = \mathcal{O}\left(1 + \sqrt{\kappa/n}\log(\Delta/\varepsilon)\right)$ and $n$ $L$-smooth and $\mu$-strongly convex functions $\{f_i : \mathbb{R}^d \to \mathbb{R}\}_{i=1}^n$ such that $f(\boldsymbol{x}_0) - f(\boldsymbol{x}^*) = \Delta$. In order to find $\hat{\boldsymbol{x}} \in \mathbb{R}^d$ such that $\mathbb{E}f(\hat{\boldsymbol{x}}) - f(\boldsymbol{x}^*) < \varepsilon$, $\mathcal{A}$ needs at least $N$ queries to $h_f$, where*

$$N = \begin{cases} \Omega\left((n+\sqrt{\kappa n})\log(\Delta/\varepsilon)\right), & for \ n = \mathcal{O}(\kappa), \\ \Omega\left(n + \left(\frac{n}{1+(\log(n/\kappa))_+}\right)\log(\Delta/\varepsilon)\right), & for \ \kappa = \mathcal{O}(n). \end{cases}$$

**Remark 3.2.** *In fact, the lower bound in Theorem 3.1 perfectly match the upper bound of the PIFO algorithm Point SAGA (Defazio, 2016)[3] in $n = \mathcal{O}(\kappa)$ case and match the the upper bound of the IFO algorithm[4] prox-SVRG (Hannah et al., 2018) in $\kappa = \mathcal{O}(n)$ case. Hence, the lower bound in Theorem 3.1 is tight, while Woodworth and Srebro (2016) only provided lower bound $\Omega\left(n+\sqrt{\kappa n}\log(1/\varepsilon)\right)$ in $n = \mathcal{O}(\kappa)$ case. The theorem also shows that the PIFO algorithm can not be more powerful than the IFO algorithm in the worst case, because Hannah et al. (2018) proposed a same lower bound for IFO algorithms.*

Next we give the lower bound when the objective function is not strongly-convex.

**Theorem 3.3.** *For any PIFO algorithm $\mathcal{A}$ and any $L, n, B, \varepsilon$ such that $\varepsilon \leq LB^2/4$, there exist a dimension $d = \mathcal{O}\left(1 + B\sqrt{L/(n\varepsilon)}\right)$ and $n$ $L$-smooth and convex functions $\{f_i : \mathbb{R}^d \to \mathbb{R}\}_{i=1}^n$ such that $\|\boldsymbol{x}_0 - \boldsymbol{x}^*\|_2 \leq B$. In order to find $\hat{\boldsymbol{x}} \in \mathbb{R}^d$ such that $\mathbb{E}f(\hat{\boldsymbol{x}}) - f(\boldsymbol{x}^*) < \varepsilon$, $\mathcal{A}$ needs at least $\Omega\left(n+B\sqrt{nL/\varepsilon}\right)$ queries to $h_f$.*

---

[3]Defazio (2016) proves Point SAGA requires $\mathcal{O}\left((n + \sqrt{\kappa n})\log(1/\varepsilon)\right)$ PIFO calls to find $\hat{\boldsymbol{x}}$ such that $\mathbb{E}\|\hat{\boldsymbol{x}} - \boldsymbol{x}^*\|_2^2 < \varepsilon\|\boldsymbol{x}_0 - \boldsymbol{x}^*\|_2^2$, which is not identical to the condition $\mathbb{E}f(\hat{\boldsymbol{x}}) - f(\boldsymbol{x}^*) < \varepsilon$ in Theorem 3.1. However, it is unnecessary to worry about it because we also establish a PIFO lower bound $\Omega\left((n + \sqrt{\kappa n})\log(1/\varepsilon)\right)$ for $\mathbb{E}\|\hat{\boldsymbol{x}} - \boldsymbol{x}^*\|_2^2 < \varepsilon\|\boldsymbol{x}_0 - \boldsymbol{x}^*\|_2^2$ in Theorem F.1.

[4]IFO algorithm is apparently also a PIFO algorithm.

**Remark 3.4.** *The lower bound in Theorem 3.3 is the same as the one of Woodworth and Srebro's result. However, our construction only requires the dimension be $\mathcal{O}\left(1 + B\sqrt{L/(n\varepsilon)}\right)$, which is much smaller than $\mathcal{O}\left(\frac{L^2 B^4}{\varepsilon^2}\log\left(\frac{nLB^2}{\varepsilon}\right)\right)$ in (Woodworth and Srebro, 2016).*

Then we extend our results to the weaker assumption: that is, the objective function $F$ is $L$-average smooth (Zhou and Gu, 2019). We start with the case that $F$ is strongly convex.

**Theorem 3.5.** *For any PIFO algorithm $\mathcal{A}$ and any $L, \mu, n, \Delta, \varepsilon$ such that $\kappa = L/\mu \geq \sqrt{3/n}\left(\frac{n}{2}+1\right)$, and $\varepsilon/\Delta \leq 0.00327$, there exist a dimension $d = \mathcal{O}\left(n^{-1/4}\sqrt{\kappa}\log\left(\Delta/\varepsilon\right)\right)$ and $n$ functions $\{f_i : \mathbb{R}^d \to \mathbb{R}\}_{i=1}^n$ where the $\{f_i\}_{i=1}^n$ are $L$-average smooth and $f$ is $\mu$-strongly convex, such that $f(\boldsymbol{x}_0) - f(\boldsymbol{x}^*) = \Delta$. In order to find $\hat{\boldsymbol{x}} \in \mathbb{R}^d$ such that $\mathbb{E}f(\hat{\boldsymbol{x}}) - f(\boldsymbol{x}^*) < \varepsilon$, $\mathcal{A}$ needs at least $\Omega\left((n+n^{3/4}\sqrt{\kappa})\log\left(\Delta/\varepsilon\right)\right)$ queries to $h_f$.*

**Remark 3.6.** *Compared with Zhou and Gu's lower bound $\Omega\left(n + n^{3/4}\sqrt{\kappa}\log\left(\Delta/\varepsilon\right)\right)$ for IFO algorithms, Theorem 3.5 shows tighter dependency on $n$ and supports PIFO algorithms additionally.*

We also give the lower bound for general convex case under the $L$-average smooth condition.

**Theorem 3.7.** *For any PIFO algorithm $\mathcal{A}$ and any $L, n, B, \varepsilon$ such that $\varepsilon \leq LB^2/4$, there exist a dimension $d = \mathcal{O}\left(1 + Bn^{-1/4}\sqrt{L/\varepsilon}\right)$ and $n$ functions $\{f_i : \mathbb{R}^d \to \mathbb{R}\}_{i=1}^n$ which the $\{f_i\}_{i=1}^n$ are $L$-average smooth and $f$ is convex, such that $\|\boldsymbol{x}_0 - \boldsymbol{x}^*\|_2 \leq B$. In order to find $\hat{\boldsymbol{x}} \in \mathbb{R}^d$ such that $\mathbb{E}f(\hat{\boldsymbol{x}}) - f(\boldsymbol{x}^*) < \varepsilon$, $\mathcal{A}$ needs at least $\Omega\left(n + Bn^{3/4}\sqrt{L/\varepsilon}\right)$ queries to $h_f$.*

**Remark 3.8.** *The lower bound in Theorem 3.7 is comparable to the one of Zhou and Gu's result, but our construction only requires the dimension be $\mathcal{O}\left(1 + Bn^{-1/4}\sqrt{L/\varepsilon}\right)$, which is much smaller than $\mathcal{O}\left(n + Bn^{3/4}\sqrt{L/\varepsilon}\right)$ in (Zhou and Gu, 2019).*

# 4 CONSTRUCTIONS IN PROOF OF MAIN THEOREMS

We demonstrate the detailed constructions for PIFO lower bounds in this section. All the omitted proof in this section can be found in Appendix for a detailed version.

## 4.1 STRONGLY CONVEX CASE

The analysis of lower bound complexity for the strongly-convex case depends on the following construction.

**Definition 4.1.** *For fixed $L, \mu, \Delta, n$, let $\alpha = \sqrt{\frac{2(L/\mu-1)}{n}+1}$. We define $f_{SC,i} : \mathbb{R}^m \to \mathbb{R}$ as follows*

$$f_{SC,i}(\boldsymbol{x}) = r_i\left(\boldsymbol{x}; \sqrt{\frac{2(L-\mu)n\Delta}{\alpha-1}}, \frac{L-\mu}{4}, \frac{\mu}{2}, m, \sqrt{\frac{2}{\alpha+1}}\right), \text{ for } 1 \leq i \leq n, \qquad (9)$$

*and*

$$F_{SC}(\boldsymbol{x}) \triangleq \frac{1}{n}\sum_{i=1}^n f_{SC,i}(\boldsymbol{x}) = \frac{L-\mu}{4n}\left\|\boldsymbol{B}\left(m, \sqrt{\frac{2}{\alpha+1}}\right)\boldsymbol{x}\right\|_2^2 + \frac{\mu}{2}\|\boldsymbol{x}\|_2^2 - \sqrt{\frac{2(L-\mu)\Delta}{n(\alpha-1)}}\langle e_m, \boldsymbol{x}\rangle.$$

Note that the $f_{SC,i}$ are $L$-smooth and $\mu$-strongly convex, and $F_{SC}(\boldsymbol{x}_0) - F_{SC}(\boldsymbol{x}^*) = \Delta$ (see Proposition E.1 in Appendix for more details). Next we show that the functions $\{f_{SC,i}\}_{i=1}^n$ are "hard enough" for any PIFO algorithm $\mathcal{A}$, and deduce the conclusion of Theorem 3.1.

**Theorem 4.2.** *Suppose that*

$$\varepsilon \leq \frac{\Delta}{9}\left(\frac{\alpha-1}{\alpha+1}\right)^2, \text{ and } m = \frac{1}{4}\left(\sqrt{2\frac{L/\mu-1}{n}+1}\right)\log\left(\frac{\Delta}{9\varepsilon}\right) + 1,$$

where $\alpha = \sqrt{\frac{2(L/\mu-1)}{n} + 1}$. *In order to find* $\hat{x} \in \mathbb{R}^m$ *such that* $\mathbb{E}F_{SC}(\hat{x}) - F_{SC}(x^*) < \varepsilon$, *PIFO algorithm* $\mathcal{A}$ *needs at least* $N$ *queries to* $h_{F_{SC}}$. *where*

$$
N = \begin{cases} \Omega\left(\left(n + \sqrt{\frac{nL}{\mu}}\right)\log\left(\frac{\Delta}{9\varepsilon}\right)\right), & for \ \frac{L}{\mu} \geq \frac{n}{2} + 1, \\ \Omega\left(n + \left(\frac{n}{1+\log(n\mu/L)}\right)\log\left(\frac{\Delta}{9\varepsilon}\right)\right), & for \ 2 \leq \frac{L}{\mu} < \frac{n}{2} + 1. \end{cases}
$$

For larger $\varepsilon$, we can apply following Lemma.

**Lemma 4.3.** *For any PIFO algorithm* $\mathcal{A}$ *and any* $L, \mu, n, \Delta, \varepsilon$ *such that* $\varepsilon \leq \Delta/2$, *there exist* $n$ *L-smooth and* $\mu$-*strongly convex functions* $\{f_i : \mathbb{R} \to \mathbb{R}\}_{i=1}^n$ *such that* $F(x_0) - F(x^*) \leq \Delta$. *In order to find* $\hat{x} \in \mathbb{R}$ *such that* $\mathbb{E}F(\hat{x}) - F(x^*) < \varepsilon$, $\mathcal{A}$ *needs at least* $\Omega(n)$ *queries to* $h_F$.

As we explain in Remark H.1, the lower bound in Lemma 4.3 is same as the lower bound in Theorem 4.2 for $\varepsilon > \frac{\Delta}{9}\left(\frac{\alpha-1}{\alpha+1}\right)^2$. In conclusion, we obtain Theorem 3.1.

## 4.2 Convex Case

The analysis of lower bound complexity for non strongly-convex cases depends on the following construction.

**Definition 4.4.** *For fixed* $L, B, n$, *we define* $f_{C,i} : \mathbb{R}^m \to \mathbb{R}$ *as follows*

$$
f_{C,i}(\boldsymbol{x}) = r_i\left(\boldsymbol{x}; \frac{\sqrt{3}}{2}\frac{BL}{(m+1)^{3/2}}, \frac{L}{4}, 0, m, 1\right) \tag{10}
$$

*and*

$$
F_C(\boldsymbol{x}) \triangleq \frac{1}{n}\sum_{i=1}^n f_{C,i}(\boldsymbol{x}) = \frac{L}{4n}\|\boldsymbol{B}(m,1)\boldsymbol{x}\|_2^2 - \frac{\sqrt{3}}{2}\frac{BL}{(m+1)^{3/2}n}\langle e_m, \boldsymbol{x}\rangle.
$$

Note that the $f_{C,i}$ are $L$-smooth and convex, and $\|\boldsymbol{x}_0 - \boldsymbol{x}^*\|_2 \leq B$ (see Proposition G.1 in Appendix for more details). Next we show the lower bound for functions $f_{C,i}$ defined above.

**Theorem 4.5.** *Suppose that*

$$
\varepsilon \leq \frac{B^2 L}{384n} \quad and \quad m = \left\lfloor \sqrt{\frac{B^2 L}{24n\varepsilon}} \right\rfloor - 1.
$$

*In order to find* $\hat{x} \in \mathbb{R}^m$ *such that* $\mathbb{E}F_C(\hat{x}) - F_C(x^*) < \varepsilon$, $\mathcal{A}$ *needs at least* $\Omega\left(n + B\sqrt{\frac{nL}{\varepsilon}}\right)$ *queries to* $h_{F_C}$.

To derive Theorem 3.3, we also need the following lemma in the case $\varepsilon > \frac{B^2 L}{384n}$.

**Lemma 4.6.** *For any PIFO algorithm* $\mathcal{A}$ *and any* $L, n, B, \varepsilon$ *such that* $\varepsilon \leq LB^2/4$, *there exist* $n$ *L-smooth and convex functions* $\{f_i : \mathbb{R} \to \mathbb{R}\}_{i=1}^n$ *such that* $|x_0 - x^*| \leq B$. *In order to find* $\hat{x} \in \mathbb{R}$ *such that* $\mathbb{E}F(\hat{x}) - F(x^*) < \varepsilon$, $\mathcal{A}$ *needs at least* $\Omega(n)$ *queries to* $h_F$.

It is worth noting that if $\varepsilon > \frac{B^2 L}{384n}$, then $\Omega(n) = \Omega\left(n + B\sqrt{\frac{nL}{\varepsilon}}\right)$. Thus combining Theorem 4.5 and Lemma 4.6, we obtain Theorem 3.3.

## 4.3 Average Smooth Case

Zhou and Gu (2019) established lower bounds of IFO complexity under the average smooth assumption. Here we demonstrate that our technique can also develop lower bounds of PIFO algorithm under this assumption.

### 4.3.1 $F$ IS STRONGLY CONVEX

For fixed $L', \mu, \Delta, n, \varepsilon$, we set $L = \sqrt{\frac{n(L'^2 - \mu^2)}{2} - \mu^2}$, and consider $\{f_{\text{SC},i}\}_{i=1}^n$ and $F_{\text{SC}}$ defined in Definition 4.1.

**Proposition 4.7.** *For $n \geq 2$, we have that*

> 1. $F_{SC}(\boldsymbol{x})$ *is $\mu$-strongly convex and $\{f_{SC,i}\}_{i=1}^n$ is $L'$-average smooth.*
>
> 2. *If $\frac{L'}{\mu} \geq \sqrt{\frac{3}{n}}(\frac{n}{2} + 1)$, then we have $\sqrt{\frac{n}{3}}L' \leq L \leq \sqrt{\frac{n}{2}}L'$ and $L/\mu \geq n/2 + 1$.*

**Theorem 4.8.** *Suppose that*

$$\frac{L'}{\mu} \geq \sqrt{\frac{3}{n}} \left( \frac{n}{2} + 1 \right), \ \varepsilon \leq \frac{\Delta}{9} \left( \frac{\sqrt{2} - 1}{\sqrt{2} + 1} \right)^2, \ and \ m = \frac{1}{4} \left( \sqrt{\sqrt{\frac{2}{n}} \frac{L'}{\mu}} + 1 \right) \log \left( \frac{\Delta}{9\varepsilon} \right) + 1.$$

*In order to find $\hat{\boldsymbol{x}} \in \mathbb{R}^m$ such that $\mathbb{E}F_{SC}(\hat{\boldsymbol{x}}) - F_{SC}(\boldsymbol{x}^*) < \varepsilon$, PIFO algorithm $\mathcal{A}$ needs at least $\Omega\left( \left( n + n^{3/4}\sqrt{\frac{L'}{\mu}} \right) \log\left( \frac{\Delta}{\varepsilon} \right) \right)$ queries to $h_{F_{SC}}$.*

### 4.3.2 $F$ IS CONVEX

For fixed $L', B, n, \varepsilon$, we set $L = \sqrt{\frac{n}{2}}L'$, and consider $\{f_{\text{C},i}\}_{i=1}^n$ and $F_{\text{C}}$ defined in Definition 4.4. It follows from Proposition 2.5 that $\{f_{\text{C},i}\}_{i=1}^n$ is $L'$-average smooth.

**Theorem 4.9.** *Suppose that*

$$\varepsilon \leq \frac{\sqrt{2}}{768} \frac{B^2 L'}{\sqrt{n}} \ and \ m = \left\lfloor \frac{\sqrt[4]{18}}{12} B n^{-1/4} \sqrt{\frac{L'}{\varepsilon}} \right\rfloor - 1.$$

*In order to find $\hat{\boldsymbol{x}} \in \mathbb{R}^m$ such that $\mathbb{E}F_C(\hat{\boldsymbol{x}}) - F_C(\boldsymbol{x}^*) < \varepsilon$, $\mathcal{A}$ needs at least $\Omega\left( n + Bn^{3/4}\sqrt{\frac{L'}{\varepsilon}} \right)$ queries to $h_{F_C}$.*

Similar to Lemma 4.6, we also need the following lemma for the case $\varepsilon > \frac{\sqrt{2}}{768} \frac{B^2 L'}{\sqrt{n}}$.

**Lemma 4.10.** *For any PIFO algorithm $\mathcal{A}$ and any $L, n, B, \varepsilon$ such that $\varepsilon \leq LB^2/4$, there exist $n$ functions $\{f_i : \mathbb{R} \to \mathbb{R}\}_{i=1}^n$ which is $L$-average smooth, such that $F(x)$ is convex and $\|x_0 - x^*\|_2 \leq B$. In order to find $\hat{x} \in \mathbb{R}$ such that $\mathbb{E}F(\hat{x}) - F(x^*) < \varepsilon$, $\mathcal{A}$ needs at least $\Omega(n)$ queries to $h_F$.*

Similarly, note that if $\varepsilon > \frac{\sqrt{2}}{768} \frac{B^2 L'}{\sqrt{n}}$, then $\Omega(n) = \Omega\left( n + Bn^{3/4}\sqrt{\frac{L'}{\varepsilon}} \right)$. In summary, we obtain Theorem 3.7.

## 5 CONCLUSION AND FUTURE WORK

In this paper we have studied lower bound of PIFO algorithm for smooth convex finite-sum optimization. We have given a tight lower bound of PIFO algorithms in the strongly convex case. We have proposed a novel construction framework that is very useful to the analysis of proximal algorithms. Based on this framework, we can extended our result to non-strongly convex, average smooth and non-convex problems easily (Appendix J). It would be interesting to prove tight lower bounds in more general setting, such as $F$ is of $(\sigma, L)$-smoothness while each $f_i$ is $(l, L)$-smoothness.

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

## A COMPARISON OF REQUIRED NUMBER OF DIMENSIONS

| | Previous Lower Bounds | Our Lower Bounds |
|---|---|---|
| $f_i$ is $L$-smooth and $\mu$-strongly convex | $\#\text{PIFO} = \Omega\left(n + \sqrt{\kappa n}\log(\frac{1}{\varepsilon})\right)$ 
 $d = \mathcal{O}\left(\frac{\kappa n}{\varepsilon}\log^5\left(\frac{1}{\varepsilon}\right)\right)$ 
 (Woodworth and Srebro, 2016) | $\#\text{PIFO} = \Omega\left((n + \sqrt{\kappa n})\log(\frac{1}{\varepsilon})\right)$ 
 $d = \mathcal{O}\left(\sqrt{\frac{\kappa}{n}}\log\left(\frac{1}{\varepsilon}\right)\right)$ 
 [Theorem 3.1] |
| $f_i$ is $L$-smooth and convex | $\#\text{PIFO} = \Omega\left(n + \sqrt{\frac{nL}{\varepsilon}}\right)$ 
 $d = \mathcal{O}\left(\frac{L^2}{\varepsilon^2}\log\left(\frac{1}{\varepsilon}\right)\right)$ 
 (Woodworth and Srebro, 2016) | $\#\text{PIFO} = \Omega\left(n + \sqrt{\frac{nL}{\varepsilon}}\right)$ 
 $d = \mathcal{O}\left(1 + \sqrt{\frac{L}{n\varepsilon}}\right)$ 
 [Theorem 3.3] |
| $\{f_i\}_{i=1}^n$ is $L$-average smooth and $f$ is $\mu$-strongly convex | $\#\text{IFO} = \Omega\left(n + n^{3/4}\sqrt{\kappa}\log\left(\frac{1}{\varepsilon}\right)\right)$ 
 $d = \mathcal{O}\left(n + n^{3/4}\sqrt{\kappa}\log\left(\frac{1}{\varepsilon}\right)\right)$ 
 (Zhou and Gu, 2019) | $\#\text{PIFO} = \Omega\left((n + n^{3/4}\sqrt{\kappa})\log\left(\frac{1}{\varepsilon}\right)\right)$ 
 $d = \mathcal{O}\left(n^{-1/4}\sqrt{\kappa}\log\left(\frac{1}{\varepsilon}\right)\right)$ 
 [Theorem 3.5] |
| $\{f_i\}_{i=1}^n$ is $L$-average smooth and $f$ is convex | $\#\text{IFO} = \Omega\left(n + n^{3/4}\sqrt{\frac{L}{\varepsilon}}\right)$ 
 $d = \mathcal{O}\left(n + n^{3/4}\sqrt{\frac{L}{\varepsilon}}\right)$ 
 (Zhou and Gu, 2019) | $\#\text{PIFO} = \Omega\left(n + n^{3/4}\sqrt{\frac{L}{\varepsilon}}\right)$ 
 $d = \mathcal{O}\left(1 + n^{-1/4}\sqrt{\frac{L}{\varepsilon}}\right)$ 
 [Theorem 3.7] |

Table 2: We compare our PIFO lower bounds with previous results, including the number of PIFO or IFO calls to obtain $\varepsilon$-suboptimal point and the required number of dimensions in corresponding construction.

## B COMPARISON WITH EXISTING PROOFS

Recall the adversary function we used is (please see detailed defintion in Section 2)

$$r(\boldsymbol{x}; \lambda_0, \lambda_1, \lambda_2, m, \omega) \triangleq \frac{1}{n}\sum_{i=1}^n r_i(\boldsymbol{x}; \lambda_0, \lambda_1, \lambda_2, m, \omega) \tag{11}$$

$$= \frac{\lambda_1}{n}\boldsymbol{x}^\top \boldsymbol{A}(m, \omega)\boldsymbol{x} + \lambda_2 \|\boldsymbol{x}\|_2^2 - \frac{\lambda_0}{n}\langle \boldsymbol{e}_m, \boldsymbol{x}\rangle, \tag{12}$$

where

$$r_i(\boldsymbol{x}; \lambda_0, \lambda_1, \lambda_2, m, \omega) = \begin{cases} \lambda_1 \sum_{l \in \mathcal{L}_1} \left\|\boldsymbol{b}_l(m, \omega)^\top \boldsymbol{x}\right\|_2^2 + \lambda_2 \|\boldsymbol{x}\|_2^2 - \lambda_0\langle \boldsymbol{e}_m, \boldsymbol{x}\rangle, & \text{for } i = 1, \\ \lambda_1 \sum_{l \in \mathcal{L}_i} \left\|\boldsymbol{b}_l(m, \omega)^\top \boldsymbol{x}\right\|_2^2 + \lambda_2 \|\boldsymbol{x}\|_2^2, & \text{for } i = 2, 3, \ldots, n. \end{cases}$$

The constructions in previous work (Lan and Zhou, 2017; Zhou and Gu, 2019) for IFO algorithms employ an aggregation of $r$ , that is,

$$f(\boldsymbol{x}) \triangleq \frac{1}{n}\sum_{i=1}^n f_i(\boldsymbol{x}), \text{ where } f_i(\boldsymbol{x}) = nr(\boldsymbol{x}_i),$$

$$\boldsymbol{x} = \begin{bmatrix} \boldsymbol{x}_1 \\ \boldsymbol{x}_2 \\ \vdots \\ \boldsymbol{x}_n \end{bmatrix} \in \mathbb{R}^{mn}, \text{ and } \boldsymbol{x}_i \in \mathbb{R}^m \text{ for } i = 1, \ldots, n.$$

The disadvantage of their construction is in that the important property from Lemma 2.6, which we can obtain information of only one extra dimension at each PIFO query, cannot be held. Note that the framework in this paper is the first lower bound analysis that utilize the decomposition form (11), which makes the "only one" property also hold for PIFO query. The previous works only consider the presentation (12) and are unaware of the decomposition (11). Moreover, the fact $r : \mathbb{R}^m \to \mathbb{R}$ and $f : \mathbb{R}^{mn} \to \mathbb{R}$ provide an intuitive understanding why our construction requires a smaller dimension (see Table 2).

The analysis in (Woodworth and Srebro, 2016; Fang et al., 2018) considers a very complicated approach to dealing with the proximal operator (completely different from how to deal with gradient operator). In contrast, our construction holds "only one" property (Lemma 2.6) both for proximal and gradient operator, which leads the proof is more concise. Our construction more clearly shows that PIFO algorithms are not more powerful than IFO algorithms in the sense of lower complexity bound. We also use our technique to prove the tight lower bound of PIFO algorithm when $\kappa = \mathcal{O}(n)$, which is a new result.

## C  Detailed Proof for Section 2

In this section, we use $\|A\|$ to denote the spectral radius of $A$.

For simplicity, let

$$
B = B(m, \omega) = \begin{bmatrix} & & & -1 & 1 \\ & & -1 & 1 & \\ & \cdot^{\cdot} & \cdot^{\cdot} & & \\ -1 & 1 & & & \\ \omega & & & & \end{bmatrix} \in \mathbb{R}^{m \times m},
$$

$b_l^\top$ is the $l$-th row of $B$, and $f_i(x) = r_i(x; \lambda_0, \lambda_1, \lambda_2, m, \omega)$.

Recall that
$$
\mathcal{L}_i = \{l : 1 \leq l \leq m, l \equiv i - 1 (\mathrm{mod} n)\}, i = 1, 2, \cdots, n.
$$
For $1 \leq i \leq n$, let $B_i$ be a submatrix which is formed from rows $\mathcal{L}_i$ of $B$, that is
$$
B_i = B[\mathcal{L}_i;]
$$

Then $f_i$ can be wriiten as

$$
f_i(x) = \lambda_1 \|B_i x\|_2^2 + \lambda_2 \|x\|_2^2 - \eta_i \langle e_m, x \rangle,
$$

where $\eta_1 = \lambda_0, \eta_i = 0, i \geq 2$.

***Proof of Proposition 2.5.***  Note that

$$
\begin{aligned}
\langle u, B_i^\top B_i u \rangle &= \|B_i u\|_2^2 \\
&= \sum_{l \in \mathcal{L}_i} (b_l^\top u)^2 \\
&= \begin{cases} \sum_{l \in \mathcal{L}_i \setminus \{m\}} (u_{m-l} - u_{m-l+1})^2 + \omega^2 u_m^2 & (\text{if } m \in \mathcal{L}_i) \\ \sum_{l \in \mathcal{L}_i} (u_{m-l} - u_{m-l+1})^2 \end{cases} \\
&\leq 2 \|u\|_2^2,
\end{aligned}
$$

where the last inequality is according to $(x+y)^2 \leq 2(x^2 + y^2)$, and $|l_1 - l_2| \geq n \geq 2$ for $l_1, l_2 \in \mathcal{L}_i$. Hence, $\|B_i^\top B_i\| \leq 2$, and

$$
\|\nabla^2 f_i(x)\| = \|2\lambda_1 B_i^\top B_i + 2\lambda_2 I\| \leq 4\lambda_1 + 2\lambda_2.
$$

Next, observe that

$$
\|\nabla f_i(x) - \nabla f_i(y)\|_2^2 = \|(2\lambda_1 B_i^\top B_i + 2\lambda_2 I)(x - y)\|_2^2
$$

Let $\boldsymbol{u} = \boldsymbol{x} - \boldsymbol{y}$.

Note that

$$\boldsymbol{b}_l \boldsymbol{b}_l^\top \boldsymbol{u} = \begin{cases} (u_{m-l} - u_{m-l+1})(\boldsymbol{e}_{m-l} - \boldsymbol{e}_{m-l+1}), & l < m, \\ \omega^2 u_1 \boldsymbol{e}_1, & l = m. \end{cases}$$

Thus, if $m \notin \mathcal{L}_i$, then

$$\left\| (2\lambda_1 \boldsymbol{B}_i^\top \boldsymbol{B}_i + 2\lambda_2 \boldsymbol{I}) \boldsymbol{u} \right\|_2^2$$

$$= \left\| 2\lambda_1 \sum_{l \in \mathcal{L}_i} (u_{m-l} - u_{m-l+1})(\boldsymbol{e}_{m-l} - \boldsymbol{e}_{m-l+1}) + 2\lambda_2 \boldsymbol{u} \right\|_2^2$$

$$= \sum_{m-l \in \mathcal{L}_i} \left[ (2\lambda_1(u_l - u_{l+1}) + 2\lambda_2 u_l)^2 + (-2\lambda_1(u_l - u_{l+1}) + 2\lambda_2 u_{l+1})^2 \right] + \sum_{\substack{m-l \notin \mathcal{L}_i \\ m-l+1 \notin \mathcal{L}_i}} (2\lambda_2 u_l)^2$$

$$\leq \sum_{m-l \in \mathcal{L}_i} 8 \left[ (\lambda_1 + \lambda_2)^2 + \lambda_1^2 \right] (u_l^2 + u_{l+1}^2) + 4\lambda_2^2 \left\| \boldsymbol{u} \right\|_2^2 .$$

Similarly, if $m \in \mathcal{L}_i$, then

$$\left\| (2\lambda_1 \boldsymbol{B}_i^\top \boldsymbol{B}_i + 2\lambda_2 \boldsymbol{I}) \boldsymbol{u} \right\|_2^2$$

$$\leq \sum_{\substack{m-l \in \mathcal{L}_i \\ l \neq 0}} 8 \left[ (\lambda_1 + \lambda_2)^2 + \lambda_1^2 \right] (u_l^2 + u_{l+1}^2) + 4(\lambda_1 \omega^2 + \lambda_2)^2 u_1^2 + 4\lambda_2^2 \left\| \boldsymbol{u} \right\|_2^2 .$$

Therefore, we have

$$\frac{1}{n} \sum_{i=1}^{n} \left\| \nabla f_i(\boldsymbol{x}) - \nabla f_i(\boldsymbol{y}) \right\|_2^2$$

$$\leq \frac{1}{n} \left[ \sum_{l=1}^{m-1} 8 \left[ (\lambda_1 + \lambda_2)^2 + \lambda_1^2 \right] (u_l^2 + u_{l+1}^2) + 4(2\lambda_1 + \lambda_2)^2 u_1^2 \right] + 4\lambda_2^2 \left\| \boldsymbol{u} \right\|_2^2$$

$$\leq \frac{16}{n} \left[ (\lambda_1 + \lambda_2)^2 + \lambda_1^2 \right] \left\| \boldsymbol{u} \right\|_2^2 + 4\lambda_2^2 \left\| \boldsymbol{u} \right\|_2^2 ,$$

where we have used $(2\lambda_1 + \lambda_2)^2 \leq 2 \left[ (\lambda_1 + \lambda_2)^2 + \lambda_1^2 \right]$.

In summary, we get that $\{f_i\}_{1 \leq i \leq n}$ is $L'$-average smooth, where

$$L' = 2\sqrt{\frac{4}{n} \left[ (\lambda_1 + \lambda_2)^2 + \lambda_1^2 \right] + \lambda_2^2}.$$

$\square$

**Proof of Lemma 2.6.** For $\boldsymbol{x} \in \mathcal{F}_k$ $(k \geq 1)$, we have

$$\boldsymbol{b}_l^\top \boldsymbol{x} = 0 \text{ for } l > k,$$
$$\boldsymbol{b}_l \in \mathcal{F}_k \text{ for } l < k,$$
$$\boldsymbol{b}_k \in \mathcal{F}_{k+1}.$$

Consequently, for $l \neq k$, $\boldsymbol{b}_l \boldsymbol{b}_l^\top \boldsymbol{x} = (\boldsymbol{b}_l^\top \boldsymbol{x}) \boldsymbol{b}_l \in \mathcal{F}_k$, and $\boldsymbol{b}_k \boldsymbol{b}_k^\top \boldsymbol{x} \in \mathcal{F}_{k+1}$.

For $k = 0$, we have $\boldsymbol{x} = \boldsymbol{0}$, and

$$\nabla f_1(\boldsymbol{x}) = \lambda_0 \boldsymbol{e}_m \in \mathcal{F}_1,$$
$$\nabla f_j(\boldsymbol{x}) = \boldsymbol{0} \ (j \geq 2).$$

Moreover, we suppose $k \geq 1$, $k \in \mathcal{L}_i$. Since

$$\nabla f_j(\boldsymbol{x}) = 2\lambda_1 \boldsymbol{B}_j^\top \boldsymbol{B}_j \boldsymbol{x} + 2\lambda_2 \boldsymbol{x} - \eta_j \boldsymbol{e}_m$$
$$= 2\lambda_1 \sum_{l \in \mathcal{L}_j} \boldsymbol{b}_l^\top \boldsymbol{b}_l \boldsymbol{x} + 2\lambda_2 \boldsymbol{x} - \eta_j \boldsymbol{e}_m.$$

Hence, $\nabla f_i(\boldsymbol{x}) \in \mathcal{F}_{k+1}$ and $\nabla f_j(\boldsymbol{x}) \in \mathcal{F}_k$ ($j \neq i$).

Now, we turn to consider $\boldsymbol{u} = \mathrm{prox}_{f_j}^\gamma(\boldsymbol{x})$. We have

$$\left(2\lambda_1 \boldsymbol{B}_j^\top \boldsymbol{B}_j + \left(2\lambda_2 + \frac{1}{\gamma}\right)\boldsymbol{I}\right) \boldsymbol{u} = \eta_j \boldsymbol{e}_m + \frac{1}{\gamma}\boldsymbol{x},$$

i.e.,

$$\boldsymbol{u} = c_1 (\boldsymbol{I} + c_2 \boldsymbol{B}_j^\top \boldsymbol{B}_j)^{-1} \boldsymbol{y},$$

where $c_1 = \frac{1}{2\lambda_2 + 1/\gamma}$, $c_2 = \frac{2\lambda_1}{2\lambda_2 + 1/\gamma}$, and $\boldsymbol{y} = \eta_j \boldsymbol{e}_m + \frac{1}{\gamma}\boldsymbol{x}$.

Note that

$$(\boldsymbol{I} + c_2 \boldsymbol{B}_j^\top \boldsymbol{B}_j)^{-1} = \boldsymbol{I} - \boldsymbol{B}_j^\top \left(\frac{1}{c_2}\boldsymbol{I} + \boldsymbol{B}_j \boldsymbol{B}_j^\top\right)^{-1} \boldsymbol{B}_j.$$

If $k = 0$ and $j > 1$, we have $\boldsymbol{y} = \boldsymbol{0}$ and $\boldsymbol{u} = \boldsymbol{0}$.
If $k = 0$ and $j = 1$, we have $\boldsymbol{y} = \lambda_0 \boldsymbol{e}_m$. On this case, $\boldsymbol{B}_1 \boldsymbol{e}_m = \boldsymbol{0}$, so $\boldsymbol{u} = c_1 \boldsymbol{y} \in \mathcal{F}_1$.

For $k \geq 1$, we know that $\boldsymbol{y} \in \mathcal{F}_k$. And observe that if $|l - l'| \geq 2$, then $\boldsymbol{b}_l^\top \boldsymbol{b}_{l'} = 0$, and consequently $\boldsymbol{B}_j \boldsymbol{B}_j^\top$ is a diagonal matrix, so we can assume that $\frac{1}{c_2}\boldsymbol{I} + \boldsymbol{B}_j \boldsymbol{B}_j^\top = \mathrm{diag}(\beta_{j,1}, \cdots, \beta_{j,|\mathcal{L}_j|})$. Therefore,

$$\boldsymbol{u} = c_1 \boldsymbol{y} - c_1 \sum_{s=1}^{|\mathcal{L}_j|} \beta_{j,s} \boldsymbol{b}_{l_{j,s}} \boldsymbol{b}_{l_{j,s}}^\top \boldsymbol{y},$$

where we assume that $\mathcal{L}_j = \{l_{j,1}, \cdots, l_{j,|\mathcal{L}_j|}\}$.

Thus, we have $\mathrm{prox}_{f_i}^\gamma(\boldsymbol{x}) \in \mathcal{F}_{k+1}$ for $k \in \mathcal{L}_i$ and $\mathrm{prox}_{f_j}^\gamma(\boldsymbol{x}) \in \mathcal{F}_k$ ($j \neq i$).

$\square$

***Proof of Corollary 2.7.*** Denote

$$\mathrm{span}\{\nabla f_{i_1}(\boldsymbol{x}_0), \cdots, \nabla f_{i_t}(\boldsymbol{x}_{t-1}), \mathrm{prox}_{f_{i_1}}^{\gamma_1}(\boldsymbol{x}_0), \cdots, \mathrm{prox}_{f_{i_t}}^{\gamma_t}(\boldsymbol{x}_{t-1})\}$$

by $\mathcal{M}_t$. We know that $\boldsymbol{x}_t \in \mathcal{M}_t$.

Suppose that $\mathcal{M}_T \subseteq \mathcal{F}_{k-1}$ for some $T$ and let $T' = \arg\min t : t > T, i_t \equiv k(\mathrm{mod}\ n)$.

By Lemma 2.6, for $T < t < T'$, we can use a simple induction to obtain that

$$\mathrm{span}\{\nabla f_{i_t}(\boldsymbol{x}_{t-1}), \mathrm{prox}_{f_{i_t}}^{\gamma_t}(\boldsymbol{x}_{t-1})\} \subseteq \mathcal{F}_{k-1}$$

and $\mathcal{M}_t \subseteq \mathcal{F}_{k-1}$.

Moreover, since $i_{T'} \equiv k(\mathrm{mod}\ n)$, we have

$$\mathrm{span}\{\nabla f_{i_{T'}}(\boldsymbol{x}_{T'-1}), \mathrm{prox}_{f_{i_{T'}}}^{\gamma_{T'}}(\boldsymbol{x}_{T'-1})\} \subseteq \mathcal{F}_k$$

and $\mathcal{M}_{T'} \subseteq \mathcal{F}_k$.

Following from above statement, it is easily to check that for $t < T_k$, we have $\boldsymbol{x}_t \in \mathcal{M}_t \subseteq \mathcal{F}_{k-1}$.

Next, note that

$$\mathbb{P}\left(T_k - T_{k-1} = s\right)$$
$$= \mathbb{P}\left(i_{T_{k-1}+1} \not\equiv k(\mathrm{mod}\ n), \cdots, i_{T_{k-1}+s-1} \not\equiv k(\mathrm{mod}\ n), i_{T_{k-1}+s} \equiv k(\mathrm{mod}\ n)\right)$$
$$= \mathbb{P}\left(i_{T_{k-1}+1} \neq k', \cdots, i_{T_{k-1}+s-1} \neq k', i_{T_{k-1}+s} = k'\right)$$
$$= (1 - p_{k'})^{s-1} p_{k'},$$

where $k' \equiv k \pmod{n}, 1 \le k' \le n$. So $T_k - T_{k-1}$ is a geometric random variable with success probability $p_{k'}$.

On the other hand, $T_k - T_{k-1}$ is just dependent on $i_{T_{k-1}+1}, \cdots, i_{T_k}$, thus for $l \neq k$, $T_l - T_{l-1}$ is independent with $T_k - T_{k-1}$.

Therefore,

$$T_k = \sum_{l=1}^{k} (T_l - T_{l-1}) = \sum_{i=1}^{k} Y_l,$$

where $Y_l$ follows a geometric distribution with success probability $q_l = p_{l'}$ where $l' \equiv l \pmod{n}, 1 \le l' \le n$.

$\square$

***Proof of Remark 2.3.*** If each $f_i$ is $L$-smooth, then for any $\boldsymbol{x}, \boldsymbol{y} \in \mathbb{R}^m$ we have

$$\|\nabla f_i(\boldsymbol{x}) - \nabla f_i(\boldsymbol{y})\|_2^2 \le L^2 \|\boldsymbol{x} - \boldsymbol{y}\|_2^2,$$

and consequently,

$$\frac{1}{n} \sum_{i=1}^{n} \|\nabla f_i(\boldsymbol{x}) - \nabla f_i(\boldsymbol{y})\|_2^2 \le L^2 \|\boldsymbol{x} - \boldsymbol{y}\|_2^2. \tag{13}$$

If $\{f_i\}_{i=1}^n$ is $L$-average smooth, then for any $\boldsymbol{x}, \boldsymbol{y} \in \mathbb{R}^m$ we have

$$
\begin{aligned}
\|\nabla f(\boldsymbol{x}) - \nabla f(\boldsymbol{y})\|_2^2 &= \frac{1}{n^2} \left\| \sum_{i=1}^{n} (\nabla f_i(\boldsymbol{x}) - \nabla f_i(\boldsymbol{y})) \right\|_2^2 \\
&\le \frac{1}{n^2} \left( \sum_{i=1}^{n} \|\nabla f_i(\boldsymbol{x}) - \nabla f_i(\boldsymbol{y})\|_2 \right)^2 \\
&\le \frac{1}{n} \sum_{i=1}^{n} \|\nabla f_i(\boldsymbol{x}) - \nabla f_i(\boldsymbol{y})\|_2^2 \\
&\le L^2 \|\boldsymbol{x} - \boldsymbol{y}\|_2^2.
\end{aligned}
$$

$\square$

***Proof of Lemma 2.9.*** Denote $\min_{\boldsymbol{x} \in \mathbb{R}^m} f(\boldsymbol{x})$ by $f^*$. For $t \le N$, we have

$$
\begin{aligned}
\mathbb{E} f(\boldsymbol{x}_t) - f^* &\ge \mathbb{E}[f(\boldsymbol{x}_t) - f^* | N < T_{M+1}] \mathbb{P}(N < T_{M+1}) \\
&\ge \mathbb{E}[\min_{\boldsymbol{x} \in \mathcal{F}_M} f(\boldsymbol{x}) - f^* | N < T_{M+1}] \mathbb{P}(N < T_{M+1}) \\
&\ge 9\varepsilon \mathbb{P}(T_{M+1} > N),
\end{aligned}
$$

where $T_{M+1}$ is defined in (7), and the second inequality follows from Corollary 2.7 (if $N < T_{M+1}$, then $\boldsymbol{x}_t \in \mathcal{F}_M$ for $t \le N$).

By Corollary 2.7, $T_{M+1}$ can be written as $T_{M+1} = \sum_{l=1}^{M+1} Y_l$, where $\{Y_l\}_{1 \le l \le M+1}$ are independent random variables, and $Y_l$ follows a geometric distribution with success probability $q_l = p_{l'}$ ($l' \equiv l \pmod{n}$, $1 \le l' \le n$). Moreover, recalling that $p_1 \le p_2 \le \cdots \le p_n$, we have $\sum_{l=1}^{M+1} q_l \le \frac{M+1}{n}$.

Therefore, by Lemma 2.8, we have

$$\mathbb{P}(T_{M+1} > N) = \mathbb{P}\left( \sum_{l=1}^{M+1} Y_l > \frac{(M+1)n}{4} \right) \ge 1 - \frac{16}{9(M+1)} \ge \frac{1}{9},$$

Hence, we can conclude that $\mathbb{E} f(\boldsymbol{x}_N) - f^* \ge 9\varepsilon \mathbb{P}(T_{M+1} > N) \ge \varepsilon$.

$\square$

**Remark**  In fact, a more strong conclusion hosts:

$$\mathbb{E}\left[\min_{t \le N} f(\boldsymbol{x}_t)\right] - \min_{\boldsymbol{x} \in \mathbb{R}^m} f(\boldsymbol{x}) \ge \varepsilon.$$

## D  RESULTS ABOUT SUM OF GEOMETRIC DISTRIBUTED RANDOM VARIABLES

**Lemma D.1.** *Let* $X_1 \sim \text{Geo}(p_1), X_2 \sim \text{Geo}(p_2)$ *be independent random variables. For any positive integer* $j$, *if* $p_1 \ne p_2$, *then*

$$\mathbb{P}\left(X_1 + X_2 > j\right) = \frac{p_2(1-p_1)^j - p_1(1-p_2)^j}{p_2 - p_1}, \tag{14}$$

*and if* $p_1 = p_2$, *then*

$$\mathbb{P}\left(X_1 + X_2 > j\right) = jp_1(1-p_1)^{j-1} + (1-p_1)^j. \tag{15}$$

*Proof.*

$$\begin{aligned}
\mathbb{P}\left(X_1 + X_2 > j\right) &= \sum_{l=1}^{j} \mathbb{P}\left(X_1 = l\right)\mathbb{P}\left(X_2 > j - l\right) + \mathbb{P}\left(X_1 > j\right) \\
&= \sum_{l=1}^{j}(1-p_1)^{l-1}p_1(1-p_2)^{j-l} + (1-p_1)^j \\
&= p_1(1-p_2)^{j-1}\sum_{l=1}^{j}\left(\frac{1-p_1}{1-p_2}\right)^{l-1} + (1-p_1)^j
\end{aligned}$$

Thus if $p_1 = p_2$, $\mathbb{P}\left(X_1 + X_2 > j\right) = jp_1(1-p_1)^{j-1} + (1-p_1)^j$.

For $p_1 \ne p_2$,

$$\begin{aligned}
\mathbb{P}\left(X_1 + X_2 > j\right) &= p_1\frac{(1-p_1)^j - (1-p_2)^j}{p_2 - p_1} + (1-p_1)^j \\
&= \frac{p_2(1-p_1)^j - p_1(1-p_2)^j}{p_2 - p_1}.
\end{aligned}$$

$\square$

**Lemma D.2.** *For* $x \ge 0$ *and* $j \ge 2$,

$$1 - \frac{j-1}{x+j/2} \le \left(\frac{x}{x+1}\right)^{j-1}. \tag{16}$$

*Proof.*  We just need to show that

$$(x+1)^{j-1}(x+j/2) - (j-1)(x+1)^{j-1} \le x^{j-1}(x+j/2),$$

that is

$$(x+1)^j - j(x+1)^{j-1}/2 - x^{j-1}(x+j/2) \le 0,$$

$$\text{i.e., } \sum_{l=0}^{j-2}\left[\binom{j}{l} - \frac{j}{2}\binom{j-1}{l}\right]x^l \le 0.$$

Note that for $l \le j - 2$,

$$\binom{j}{l} - \frac{j}{2}\binom{j-1}{l} = \left(1 - \frac{j-l}{2}\right)\binom{j}{l} \le 0,$$

thus inequality (16) hosts for $x \ge 0$ and $j \ge 2$.

$\square$

**Lemma D.3.** *Let $X_1 \sim \text{Geo}(p_1), X_2 \sim \text{Geo}(p_2), Y_1, Y_2 \sim \text{Geo}\left(\frac{p_1+p_2}{2}\right)$ be independent random variables with $0 < p_1 \leq p_2 \leq 1$. Then for any positive integer $j$, we have*

$$\mathbb{P}\left(X_1 + X_2 > j\right) \geq \mathbb{P}\left(Y_1 + Y_2 > j\right).$$

*Proof.* If $j = 1$, then $\mathbb{P}\left(X_1 + X_2 > j\right) = 1 = \mathbb{P}\left(Y_1 + Y_2 > j\right)$.
If $p_1 = p_2 = 1$, then $\mathbb{P}\left(X_1 + X_2 > j\right) = 0 = \mathbb{P}\left(Y_1 + Y_2 > j\right)$ for $j \geq 2$.

Let $j \geq 2$, and $c \triangleq p_1 + p_2 < 2$ be a given constant.

We prove that $f(p_1) \triangleq \mathbb{P}\left(X_1 + X_2 > j\right)$ is a decreasing function.

Employing equation (14), for $p_1 < c/2$, we have

$$f(p_1) = \frac{(c - p_1)(1 - p_1)^j - p_1(1 + p_1 - c)^j}{c - 2p_1},$$

and

$$f'(p_1) = \frac{-(1 - p_1)^j - j(c - p_1)(1 - p_1)^{j-1} - (1 + p_1 - c)^j - jp_1(1 + p_1 - c)^{j-1}}{c - 2p_1}$$

$$+ 2\frac{(c - p_1)(1 - p_1)^j - p_1(1 + p_1 - c)^j}{(c - 2p_1)^2}$$

$$= \frac{[c(1 - p_1) - j(c - p_1)(c - 2p_1)](1 - p_1)^{j-1} - [c(1 + p_1 - c) + jp_1(c - 2p_1)](1 + p_1 - c)^{j-1}}{(c - 2p_1)^2}.$$

Hence $f'(p_1) < 0$ is equivalent to

$$\frac{c(1 - p_1) - j(c - p_1)(c - 2p_1)}{c(1 + p_1 - c) + jp_1(c - 2p_1)} < \left(\frac{1 + p_1 - c}{1 - p_1}\right)^{j-1}. \tag{17}$$

Note that

$$\frac{c(1 - p_1) - j(c - p_1)(c - 2p_1)}{c(1 + p_1 - c) + jp_1(c - 2p_1)}$$

$$= 1 - \frac{(j - 1)c(c - 2p_1)}{c(1 + p_1 - c) + jp_1(c - 2p_1)}$$

$$= 1 - \frac{j - 1}{\frac{1 + p_1 - c}{c - 2p_1} + j\frac{p_1}{c}}$$

Denote $x = \frac{1 + p_1 - c}{c - 2p_1}$. If $c \leq 1$, then $p_1 > 0$ and $x > \frac{1-c}{c} \geq 0$. And if $c > 1$, then $p_1 \geq c - 1$ and $x \geq \frac{1+c-1-c}{2-c} = 0$.
Rewrite inequality (17) as

$$1 - \frac{j - 1}{x + jp_1/c} < \left(\frac{x}{x + 1}\right)^{j-1}.$$

Recall inequality (16), we have

$$\left(\frac{x}{x + 1}\right)^{j-1} \geq 1 - \frac{j - 1}{x + j/2} > 1 - \frac{j - 1}{x + jp_1/c}.$$

Consequently, $f'(p_1) < 0$ hosts for $p_1 < c/2$ and $j \geq 2$.
With the fact that $\lim_{p_1 \to c/2} f(p_1) = f(c/2)$ according to equation (15), we have

$$\mathbb{P}\left(X_1 + X_2 > j\right) \geq \mathbb{P}\left(Y_1 + Y_2 > j\right).$$

for any positive integer $j$ and $0 < p_1 \leq p_2 \leq 1$. $\qquad\square$

**Corollary D.4.** *Let $X_1 \sim \mathrm{Geo}(p_1), X_2 \sim \mathrm{Geo}(p_2), Y_1, Y_2 \sim \mathrm{Geo}\left(\frac{p_1+p_2}{2}\right)$ be independent random variables with $0 < p_1 \leq p_2 \leq 1$. Suppose $Z$ is a random variable that takes nonnegative integer values, and $Z$ is independent with $X_1, X_2, Y_1, Y_2$. Then for any positive integer $j$, we have*

$$\mathbb{P}\left(Z + X_1 + X_2 > j\right) \geq \mathbb{P}\left(Z + Y_1 + Y_2 > j\right).$$

*Proof.* With applying Lemma D.3, we have

$$
\begin{aligned}
\mathbb{P}\left(Z + X_1 + X_2 > j\right) &= \sum_{l=0}^{j-1} \mathbb{P}\left(Z = l\right) \mathbb{P}\left(X_1 + X_2 > l - j\right) + \mathbb{P}\left(Z > j - 1\right) \\
&\geq \sum_{l=0}^{j-1} \mathbb{P}\left(Z = l\right) \mathbb{P}\left(Y_1 + Y_2 > l - j\right) + \mathbb{P}\left(Z > j - 1\right) \\
&= \mathbb{P}\left(Z + Y_1 + Y_2 > j\right).
\end{aligned}
$$

$\square$

**Corollary D.5.** *Let $\{X_i\}_{1 \leq i \leq m}$ be independent variables, and $X_i$ follow a geometric distribution with success probability $p_i$. For any positive integer $j$, we have*

$$\mathbb{P}\left(\sum_{i=1}^{m} X_i \geq j\right) \geq \mathbb{P}\left(\sum_{i=1}^{m} Y_i \geq j\right),$$

*where $\{Y_i\}_{1 \leq i \leq m}$ are i.i.d. random variables, $Y_i \sim \mathrm{Geo}(\sum_{i=1}^{m} p_i/m)$, and $Y_i$ is independent with $X_{i'} (1 \leq i' \leq m)$.*

*Proof.* Let

$$f(p_1, p_2, \cdots, p_m) \triangleq \mathbb{P}\left(\sum_{i=1}^{m} X_i \geq j\right).$$

Our goal is to minimize $f(p_1, p_2, \cdots, p_m)$ such that $\sum_{i=1}^{m} p_i = S < 1$.

By Corollary D.4, we know that

$$f(p_1, p_2, \cdots, p_i, \cdots, p_j, \cdots, p_m) \geq f(p_1, p_2, \cdots, \frac{p_i + p_j}{2}, \cdots, \frac{p_i + p_j}{2}, \cdots, p_m).$$

This fact implies that $(p_1, p_2, \cdots, p_m)$ such that $p_1 = p_2 = \cdots = p_m = S/m$ is a minimizer of the function $f$.

$\square$

**Lemma D.6.** *Let $\{X_i\}_{1 \leq i \leq m}$ be i.i.d. random variables, and $X_i$ follows a geometric distribution with success probability $p$. We have*

$$\mathbb{P}\left(\sum_{i=1}^{m} X_i > \frac{m}{4p}\right) \geq 1 - \frac{16}{9m} \tag{18}$$

*Proof.* Denote $\sum_{i=1}^{m} X_i$ by $\tau$. We know that

$$\mathbb{E}\tau = \frac{m}{p}, \quad \mathrm{Var}(\tau) = \frac{m(1-p)}{p^2}.$$

Hence, we have

$$
\begin{aligned}
\mathbb{P}\left(\tau > \frac{1}{4}\mathbb{E}\tau\right) &= \mathbb{P}\left(\tau - \mathbb{E}\tau > -\frac{3}{4}\mathbb{E}\tau\right) \\
&= 1 - \mathbb{P}\left(\tau - \mathbb{E}\tau \leq -\frac{3}{4}\mathbb{E}\tau\right) \\
&\geq 1 - \mathbb{P}\left(|\tau - \mathbb{E}\tau| \geq \frac{3}{4}\mathbb{E}\tau\right) \\
&\geq 1 - \frac{16\mathrm{Var}(\tau)}{9(\mathbb{E}\tau)^2} \\
&= 1 - \frac{16m(1-p)}{9m^2} \geq 1 - \frac{16}{9m}.
\end{aligned}
$$

$\square$

**Corollary D.7.** *Let $\{X_i\}_{1 \leq i \leq m}$ be independent random variables, and $X_i$ follows a geometric distribution with success probability $p_i$. Then*

$$
\mathbb{P}\left(\sum_{i=1}^{m} X_i > \frac{m^2}{4(\sum_{i=1}^{m} p_i)}\right) \geq 1 - \frac{16}{9m}.
$$

## E  PROOF OF THEOREM 4.2

**Proposition E.1.** *For any $n \geq 2$, $m \geq 2$, $f_{SC,i}$ and $F_{SC}$ in Definition 4.1 satisfy:*

1. *$f_{SC,i}$ is $L$-smooth and $\mu$-strongly convex.*

2. *The minimizer of the function $F_{SC}$ is*

$$
\boldsymbol{x}^* = \arg\min_{\boldsymbol{x} \in \mathbb{R}^m} F_{SC}(\boldsymbol{x}) = \sqrt{\frac{2\Delta n(\alpha+1)^2}{(L-\mu)(\alpha-1)}}(q^m, q^{m-1}, \cdots, q)^\top,
$$

   *where $q = \frac{\alpha-1}{\alpha+1}$. Moreover, $F_{SC}(\boldsymbol{x}^*) = -\Delta$.*

3. *For $1 \leq k \leq m-1$, we have*

$$
\min_{\boldsymbol{x} \in \mathcal{F}_k} F_{SC}(\boldsymbol{x}) - F_{SC}(\boldsymbol{x}^*) \geq \Delta q^{2k}. \tag{19}
$$

*Proof.*

1. Just recall Proposition 2.5.

2. Denote $\xi = \sqrt{\frac{2\Delta n(\alpha+1)^2}{(L-\mu)(\alpha-1)}}$.

   Let $\nabla F_{SC}(\boldsymbol{x}) = 0$, that is

$$
\left(\frac{L-\mu}{2n}\boldsymbol{A}\left(\sqrt{\frac{2}{\alpha+1}}\right) + \mu\boldsymbol{I}\right)\boldsymbol{x} = \frac{L-\mu}{n(\alpha+1)}\xi\boldsymbol{e}_m,
$$

   or

$$
\begin{bmatrix}
\omega^2 + 1 + \frac{2n\mu}{L-\mu} & -1 & & & \\
-1 & 2 + \frac{2n\mu}{L-\mu} & -1 & & \\
& \ddots & \ddots & & \\
& & -1 & 2 + \frac{2n\mu}{L-\mu} & -1 \\
& & & -1 & 1 + \frac{2n\mu}{L-\mu}
\end{bmatrix}\boldsymbol{x} = \begin{bmatrix} 0 \\ 0 \\ \vdots \\ 0 \\ \frac{2\xi}{\alpha+1} \end{bmatrix} \tag{20}
$$

Note that $q = \frac{\alpha-1}{\alpha+1}$ is a root of the equation

$$z^2 - \left(2 + \frac{2n\mu}{L-\mu}\right)z + 1 = 0,$$

and

$$\omega^2 + 1 + \frac{2n\mu}{L-\mu} = \frac{1}{q},$$

$$\frac{2}{\alpha+1} = 1 - q = -q^2 + (1 + \frac{2n\mu}{L-\mu})q.$$

Hence, it is easily to check that the solution to Equation (20) is

$$\boldsymbol{x}^* = \xi(q^m, q^{m-1}, \cdots, q)^\top,$$

and

$$F_{\text{SC}}(\boldsymbol{x}^*) = -\frac{L-\mu}{2n(\alpha+1)}\xi^2 q = -\Delta.$$

3. If $\boldsymbol{x} \in \mathcal{F}_k$, $1 \le k < m$, then $x_1 = x_2 = \cdots = x_{m-k} = 0$.

Let $\boldsymbol{y} = \boldsymbol{x}_{m-k+1:m} \in \mathbb{R}^k$ and $\boldsymbol{A}_k$ be last $k$ rows and columns of the matrix in Equation (21). Then we can rewrite $F(\boldsymbol{x})$ as

$$F_k(\boldsymbol{y}) \triangleq F_{\text{SC}}(\boldsymbol{x}) = \frac{L-\mu}{4n}\boldsymbol{y}^\top \boldsymbol{A}_k \boldsymbol{y} - \frac{L-\mu}{n(\alpha+1)}\xi\langle \boldsymbol{e}_m, \boldsymbol{y}\rangle.$$

Let $\nabla F_k(\boldsymbol{y}) = 0$, that is

$$\begin{bmatrix} 2 + \frac{2n\mu}{L-\mu} & -1 & & & \\ -1 & 2 + \frac{2n\mu}{L-\mu} & -1 & & \\ & \ddots & \ddots & & \\ & & -1 & 2 + \frac{2n\mu}{L-\mu} & -1 \\ & & & -1 & 1 + \frac{2n\mu}{L-\mu} \end{bmatrix} \boldsymbol{y} = \begin{bmatrix} 0 \\ 0 \\ \vdots \\ 0 \\ \frac{2\xi}{\alpha+1} \end{bmatrix}. \qquad (21)$$

By some calculation, the solution to above equation is

$$\frac{\xi q^{k+1}}{1+q^{2k+1}}\left(q^{-1} - q, q^{-2} - q^2, \cdots, q^{-k} - q^k\right)^\top.$$

Thus

$$\min_{\boldsymbol{x}\in\mathcal{F}_k} F_{\text{SC}}(\boldsymbol{x}) = \min_{\boldsymbol{y}\in\mathbb{R}^k} F_k(\boldsymbol{y}) = -\frac{L-\mu}{2n(\alpha+1)}\xi^2 q \frac{1-q^{2k}}{1+q^{2k+1}} = \Delta\frac{1-q^{2k}}{1+q^{2k+1}},$$

and

$$\min_{\boldsymbol{x}\in\mathcal{F}_k} F_{\text{SC}}(\boldsymbol{x}) - F_{\text{SC}}(\boldsymbol{x}^*) = \Delta\left(1 - \frac{1-q^{2k}}{1+q^{2k+1}}\right)$$

$$= \Delta q^{2k}\frac{1+q}{1+q^{2k+1}} \ge \Delta q^{2k}.$$

$\square$

***Proof of Theorem 4.2.*** Let $M = \left\lfloor \frac{\log(9\varepsilon/\Delta)}{2\log q} \right\rfloor$, then we have

$$\arg\min_{\boldsymbol{x}\in\mathcal{F}_M} F_{\text{SC}}(\boldsymbol{x}) - F_{\text{SC}}(\boldsymbol{x}^*) \ge \Delta q^{2M} \ge 9\varepsilon,$$

where the first inequality is according to the third property of Proposition E.1.

Following from Lemma 2.9, for $M \geq 1$ and $N = (M+1)n/4$, we have

$$\min_{t \leq N} \mathbb{E} F_{\text{SC}}(\boldsymbol{x}_t) - F_{\text{SC}}(\boldsymbol{x}^*) \geq \varepsilon.$$

Therefore, in order to find $\hat{\boldsymbol{x}} \in \mathbb{R}^m$ such that $\mathbb{E} F_{\text{SC}}(\hat{\boldsymbol{x}}) - F_{\text{SC}}(\boldsymbol{x}^*) < \varepsilon$, $\mathcal{A}$ needs at least $N$ queries to $h_{F_{\text{SC}}}$.

We estimate $-\log(q)$ and $N$ in two cases.

1. If $L/\mu \geq n/2 + 1$, then $\alpha = \sqrt{2\frac{L/\mu-1}{n} + 1} \geq \sqrt{2}$. Observe that function $h(\beta) = \frac{1}{\log\left(\frac{\beta+1}{\beta-1}\right)} - \frac{\beta}{2}$ is increasing when $\beta > 1$. Thus, we have

$$-\frac{1}{\log(q)} = \frac{1}{\log\left(\frac{\alpha+1}{\alpha-1}\right)} \geq \frac{\alpha}{2} + h(\sqrt{2})$$

$$= \frac{1}{2}\sqrt{2\frac{L/\mu-1}{n} + 1} + h(\sqrt{2})$$

$$\geq \frac{\sqrt{2}}{4}\left(\sqrt{2\frac{L/\mu-1}{n}} + 1\right) + h(\sqrt{2})$$

$$\geq \frac{1}{2}\sqrt{\frac{L/\mu-1}{n}} + \frac{\sqrt{2}}{4} + h(\sqrt{2}),$$

and

$$N = (M+1)n/4 = \frac{n}{4}\left(\left\lfloor \frac{\log(9\varepsilon/\Delta)}{2\log q} \right\rfloor + 1\right)$$

$$\geq \frac{n}{8}\left(-\frac{1}{\log(q)}\right)\log\left(\frac{\Delta}{9\varepsilon}\right)$$

$$\geq \frac{n}{8}\left(\frac{1}{2}\sqrt{\frac{L/\mu-1}{n}} + \frac{\sqrt{2}}{4} + h(\sqrt{2})\right)\log\left(\frac{\Delta}{9\varepsilon}\right)$$

$$= \Omega\left(\left(n + \sqrt{\frac{nL}{\mu}}\right)\log\left(\frac{\Delta}{9\varepsilon}\right)\right)$$

2. If $2 \leq L/\mu < n/2 + 1$, then we have

$$-\log(q) = \log\left(\frac{\alpha+1}{\alpha-1}\right) = \log\left(1 + \frac{2(\alpha-1)}{\alpha^2-1}\right)$$

$$= \log\left(1 + \frac{\sqrt{2\frac{L/\mu-1}{n}+1} - 1}{\frac{L/\mu-1}{n}}\right) \leq \log\left(1 + \frac{(\sqrt{2}-1)n}{L/\mu-1}\right)$$

$$\leq \log\left(\frac{(\sqrt{2}-1/2)n}{L/\mu-1}\right) \leq \log\left(\frac{(2\sqrt{2}-1)n}{L/\mu}\right), \tag{22}$$

where the first inequality and second inequality follow from $L/\mu - 1 < n/2$ and the last inequality is according to $\frac{1}{x-1} \leq \frac{2}{x}$ for $x \geq 2$.

Note that $n \geq 2$, thus $\frac{n}{n-1} \leq 2 \leq \frac{n}{L/\mu-1}$, and hence $n \geq L/\mu$, i.e. $\log(n\mu/L) \geq 0$.

Therefore,

$$N = (M+1)n/4 \geq \frac{n}{8}\left(-\frac{1}{\log(q)}\right)\log\left(\frac{\Delta}{9\varepsilon}\right)$$

$$= \Omega\left(\left(\frac{n}{1 + \log(n\mu/L)}\right)\log\left(\frac{\Delta}{9\varepsilon}\right)\right).$$

Recalling that we assume that $9\varepsilon/\Delta \le q^2$, thus we have

$$N \ge \frac{n}{8}\left(-\frac{1}{\log(q)}\right)\log\left(\frac{\Delta}{9\varepsilon}\right) \ge \frac{n}{8}\left(-\frac{1}{\log(q)}\right)(-2\log(q)) = \frac{n}{4}.$$

Therefore, $N = \Omega\left(n + \left(\frac{n}{1+\log(n\mu/L)}\right)\log\left(\frac{\Delta}{9\varepsilon}\right)\right)$.

At last, we must to ensure that $1 \le M < m$, that is

$$1 \le \frac{\log(9\varepsilon/\Delta)}{2\log q} < m. \tag{23}$$

Note that $\lim_{\beta \to +\infty} h(\beta) = 0$, so $-1/\log(q) \le \alpha/2$. Thus the above conditions are satisfied when

$$m = \frac{\log(\Delta/(9\varepsilon))}{2(-\log q)} + 1 \le \frac{1}{4}\left(\sqrt{2\frac{L/\mu - 1}{n}} + 1\right)\log\left(\frac{\Delta}{9\varepsilon}\right) + 1 = \mathcal{O}\left(\sqrt{\frac{L}{n\mu}}\log\left(\frac{\Delta}{\varepsilon}\right)\right),$$

and

$$\frac{\varepsilon}{\Delta} \le \frac{1}{9}\left(\frac{\alpha - 1}{\alpha + 1}\right)^2.$$

$\square$

## F    LOWER BOUND FOR ANOTHER FORM OF SUBOPTIMAL SOLUTION

Defazio (2016) showed that the PIFO algorithm Point SAGA has the convergence result $\mathbb{E}\|\boldsymbol{x}_t - \boldsymbol{x}^*\|_2^2 \le (q')^t\|\boldsymbol{x}_0 - \boldsymbol{x}^*\|_2$, where $q'$ satisfies $-1/\log(q') = \mathcal{O}\left(n + \sqrt{nL/\mu}\right)$. To match this form of upper bound, we point out that a similar result holds for $\{f_{\mathrm{SC},i}\}_{i=1}^n$.

**Theorem F.1.** *Suppose that*

$$\frac{L}{\mu} \ge \frac{n}{2} + 1, \ \varepsilon \le \frac{1}{18}\left(\frac{\sqrt{2}-1}{\sqrt{2}+1}\right)^2, \ \textit{and } m = \frac{1}{2}\left(\sqrt{2\frac{L/\mu - 1}{n}} + 1\right)\log\left(\frac{1}{18\varepsilon}\right) + 1.$$

*In order to find $\hat{\boldsymbol{x}} \in \mathbb{R}^m$ such that $\mathbb{E}\|\hat{\boldsymbol{x}} - \boldsymbol{x}^*\|_2^2 < \varepsilon\|\boldsymbol{x}_0 - \boldsymbol{x}^*\|_2^2$, PIFO algorithm $\mathcal{A}$ needs at least $\Omega\left(\left(n + \sqrt{\frac{nL}{\mu}}\right)\log\left(\frac{1}{\varepsilon}\right)\right)$ queries to $h_{F_{SC}}$.*

*Proof.* Denote $\xi = \sqrt{\frac{2\Delta n(\alpha+1)^2}{(L-\mu)(\alpha-1)}}$, and $M = \left\lfloor\frac{\log(18\varepsilon)}{2\log q}\right\rfloor$.

For $1 \le M \le m/2$, $N = n(M+1)/4$ and $t \le N$, we have

$$\mathbb{E}\|\boldsymbol{x}_t - \boldsymbol{x}^*\|_2^2 \ge \mathbb{E}\left[\|\boldsymbol{x}_t - \boldsymbol{x}^*\|_2^2 \,\Big|\, N < T_{M+1}\right]\mathbb{P}(N < T_{M+1})$$

$$\ge \mathbb{E}\left[\min_{\boldsymbol{x} \in \mathcal{F}_M}\|\boldsymbol{x} - \boldsymbol{x}^*\|_2^2 \,\Big|\, N < T_{M+1}\right]\mathbb{P}(N < T_{M+1})$$

$$\ge \frac{1}{9}\min_{\boldsymbol{x} \in \mathcal{F}_M}\|\boldsymbol{x} - \boldsymbol{x}^*\|_2^2.$$

where $T_{M+1}$ is defined in (7), the second inequality follows from Corollary 2.7 (if $N < T_{M+1}$, then $\boldsymbol{x}_t \in \mathcal{F}_M$ for $t \le N$), and the last inequality is established because of our Corollary 2.7 (More detailed explanation refer to our proof of Lemma 2.9).

By Proposition E.1, we know that $\boldsymbol{x}^* = \xi(q^m, q^{m-1}, \cdots, q)^\top$, and

$$\|\boldsymbol{x}_0 - \boldsymbol{x}^*\|_2^2 = \|\boldsymbol{x}^*\|_2^2 = \xi^2\frac{q^2 - q^{2(m+1)}}{1 - q^2}.$$

Note that if $x \in \mathcal{F}_M$, then $x_1 = x_2 = \cdots = x_{m-M} = 0$, thus

$$\min_{x \in \mathcal{F}_M} \|x - x^*\|_2^2 = \xi^2 \sum_{l=m-M}^{m} q^{2(m-l+1)} = \xi^2 \frac{q^{2(M+1)} - q^{2(m+1)}}{1 - q^2}.$$

Thus, for $t \le N$ and $M \le m/2$, we have

$$\frac{\mathbb{E}\|x_t - x^*\|_2^2}{\|x_t - x^*\|_2^2} \ge \frac{1}{9} \frac{q^{2M} - q^{2m}}{1 - q^{2m}}$$

$$\ge \frac{1}{18} q^{2M} = \frac{1}{18} q^{2\left\lfloor \frac{\log(18\varepsilon)}{2\log q} \right\rfloor} \ge \varepsilon,$$

where the second inequality is due to

$$\frac{q^{2M} - q^{2m}}{1 - q^{2m}} - \frac{q^{2M}}{2} = \frac{q^{2M} - 2q^{2m} + q^{2(m+M)}}{2(1 - q^{2m})}$$

$$= \frac{q^{2M}}{2(1 - q^{2m})} (1 - 2q^{2(m-M)} + q^{2m})$$

$$\ge \frac{q^{2M}}{2(1 - q^{2m})} (1 - 2q^m + q^{2m}) \ge 0.$$

Therefore, in order to find $\hat{x} \in \mathbb{R}^m$ such that $\frac{\mathbb{E}\|\hat{x} - x^*\|_2^2}{\|x_0 - x^*\|_2^2} < \varepsilon$, $\mathcal{A}$ needs at least $N$ queries to $h_{F_{\mathrm{sc}}}$.

As we have showed in proof of Theorem 4.2, for $L/\mu \ge n/2 + 1$, we have

$$\frac{1}{2} \sqrt{2 \frac{L/\mu - 1}{n} + 1} \ge -\frac{1}{\log(q)} \ge c_1 \left( \sqrt{\frac{L/\mu - 1}{n}} + 1 \right),$$

and

$$N = \frac{n}{4}(M + 1) \ge \frac{n}{4} \frac{\log(18\varepsilon)}{2\log q}$$

$$\ge \frac{c_1}{8} \left( n + \sqrt{n(L/\mu - 1)} \right) \log \left( \frac{1}{18\varepsilon} \right)$$

$$= \Omega \left( \left( n + \sqrt{\frac{nL}{\mu}} \right) \log \left( \frac{1}{\varepsilon} \right) \right).$$

At last, we have to ensure that $1 \le M \le m/2$, that is

$$1 \le \frac{\log(18\varepsilon)}{2\log q} < m/2.$$

The above conditions are satisfied when

$$m = \frac{\log(1/(18\varepsilon))}{-\log q} + 1 \le \frac{1}{2} \left( \sqrt{2 \frac{L/\mu - 1}{n}} + 1 \right) \log \left( \frac{1}{18\varepsilon} \right) + 1 = \mathcal{O} \left( \sqrt{\frac{L}{n\mu}} \log \left( \frac{1}{\varepsilon} \right) \right),$$

and

$$\varepsilon \le \frac{1}{18} q^2.$$

Observe that when $L/\mu \le n/2 + 1$, we have $\alpha \ge \sqrt{2}$ and $q = \frac{\alpha - 1}{\alpha + 1} \ge \frac{\sqrt{2} - 1}{\sqrt{2} + 1}$. Hence, we just need $\varepsilon \le \frac{1}{18} \left( \frac{\sqrt{2} - 1}{\sqrt{2} + 1} \right)^2 \approx 0.00164$.

$\square$

## G    PROOF OF THEOREM 4.5

**Proposition G.1.** *For any $n \geq 2$, $m \geq 2$, following properties hold:*

1. *$f_{C,i}$ is L-smooth and convex.*

2. *The minimizer of the function $F_C$ is*

$$\boldsymbol{x}^* = \arg\min_{\boldsymbol{x} \in \mathbb{R}^m} F_C(\boldsymbol{x}) = \frac{2\xi}{L} (1, 2, \cdots, m)^\top,$$

   *where $\xi = \frac{\sqrt{3}}{2} \frac{BL}{(m+1)^{3/2}}$. Moreover, $F_C(\boldsymbol{x}^*) = -\frac{m\xi^2}{nL}$ and $\|\boldsymbol{x}_0 - \boldsymbol{x}^*\|_2^2 \leq B^2$.*

3. *For $1 \leq k \leq m$, we have*

$$\min_{\boldsymbol{x} \in \mathcal{F}_k} F_C(\boldsymbol{x}) - F_C(\boldsymbol{x}^*) = \frac{\xi^2}{nL}(m - k). \tag{24}$$

*Proof.*

1. Just recall Proposition 2.5.

2. Denote $\xi = \frac{\sqrt{3}}{2} \frac{BL}{(m+1)^{3/2}n}$. Let $\nabla F_C(\boldsymbol{x}) = 0$, that is

$$\frac{L}{2n} \boldsymbol{A}(1)\boldsymbol{x} = \frac{\xi}{n} \boldsymbol{e}_m,$$

   or

$$\begin{bmatrix} 2 & -1 & & & \\ -1 & 2 & -1 & & \\ & \ddots & \ddots & & \\ & & -1 & 2 & -1 \\ & & & -1 & 1 \end{bmatrix} \boldsymbol{x} = \begin{bmatrix} 0 \\ 0 \\ \vdots \\ 0 \\ \frac{2\xi}{L} \end{bmatrix}. \tag{25}$$

   Hence, it is easily to check that the solution to Equation (25) is

$$\boldsymbol{x}^* = \frac{2\xi}{L}(1, 2, \cdots, m)^\top,$$

   and

$$F_C(\boldsymbol{x}^*) = -\frac{m\xi^2}{nL}.$$

   Moreover, we have

$$\begin{aligned} \|\boldsymbol{x}_0 - \boldsymbol{x}^*\|_2^2 &= \frac{4\xi^2}{L^2} \frac{m(m+1)(2m+1)}{6} \\ &\leq \frac{4\xi^2}{3L^2}(m+1)^3 = B^2. \end{aligned}$$

3. By similar calculation to above proof, we have

$$\arg\min_{\boldsymbol{x} \in \mathcal{F}_k} F_C(\boldsymbol{x}) = \frac{2\xi}{L}(1, 2, \cdots, k)^\top,$$

   and

$$\min_{\boldsymbol{x} \in \mathcal{F}_k} F_C(\boldsymbol{x}) = -\frac{k\xi^2}{nL}.$$

   Thus

$$\min_{\boldsymbol{x} \in \mathcal{F}_k} F_C(\boldsymbol{x}) - F_C(\boldsymbol{x}^*) = \frac{\xi^2}{nL}(m - k).$$

$\square$

**Proof of Theorem 4.5.** Since $\varepsilon \leq \frac{B^2 L}{384n}$, we have $m \geq 3$. Let $\xi = \frac{\sqrt{3}}{2} \frac{BL}{(m+1)^{3/2}}$.

For $M = \lfloor \frac{m-1}{2} \rfloor \geq 1$, we have $m - M \geq (m+1)/2$, and

$$\min_{\boldsymbol{x} \in \mathcal{F}_M} F_{\mathrm{C}}(\boldsymbol{x}) - F_{\mathrm{C}}(\boldsymbol{x}^*) = \frac{\xi^2}{nL}(m - M) = \frac{3B^2 L}{4n} \frac{m - M}{(m+1)^3}$$

$$\geq \frac{3B^2 L}{8n} \frac{1}{(m+1)^2} \geq 9\varepsilon,$$

where the first equation is according to the 3rd property in Proposition G.1 and the last inequality follows from $m + 1 \leq B\sqrt{L/(24n\varepsilon)}$.

Similar to the proof of Theorem 4.2, by Lemma 2.9, we have

$$\min_{t \leq N} \mathbb{E} F_{\mathrm{C}}(\boldsymbol{x}_t) - F_{\mathrm{C}}(\boldsymbol{x}^*) \geq \varepsilon.$$

In other words, in order to find $\hat{\boldsymbol{x}} \in \mathbb{R}^m$ such that $\mathbb{E} F_{\mathrm{C}}(\hat{\boldsymbol{x}}) - F_{\mathrm{C}}(\boldsymbol{x}^*) < \varepsilon$, $\mathcal{A}$ needs at least $N$ queries to $h_F$.

At last, observe that

$$N = (M + 1)n/4 = \frac{n}{4} \left\lfloor \frac{m+1}{2} \right\rfloor$$

$$\geq \frac{n(m-1)}{8}$$

$$\geq \frac{n}{8} \left( \sqrt{\frac{B^2 L}{24n\varepsilon}} - 2 \right)$$

$$= \Omega \left( n + B\sqrt{\frac{nL}{\varepsilon}} \right),$$

where we have recalled $\varepsilon \leq \frac{B^2 L}{384n}$ in last equation.

$\square$

## H    PROOF OF LEMMA 4.3, LEMMA 4.6 AND LEMMA 4.10

**Proof of Lemma 4.6.** Consider the following functions $\{g_i\}_{1 \leq i \leq n}$, $g_i : \mathbb{R} \to \mathbb{R}$, where

$$g_1(x) = \frac{L}{2}x^2 - nLBx,$$

$$g_i(x) = \frac{L}{2}x^2,$$

$$G(x) = \frac{1}{n} \sum_{i=1}^{n} g_i(x) = \frac{L}{2}x^2 - LBx.$$

First observe that

$$x^* = \arg\min_{x \in \mathbb{R}} G(x) = B,$$

$$G(0) - G(x^*) = \frac{LB^2}{2},$$

and $|x_0 - x^*| = B$.

For $i > 1$, we have $\frac{dg_i(x)}{dx}|_{x=0} = 0$ and $\mathrm{prox}_{g_i}^{\gamma}(0) = 0$. Thus $x_t = 0$ will host till our first-order method $\mathcal{A}$ draws the component $f_1$. That is, for $t < T = \arg\min\{t : i_t = 1\}$, we have $x_t = 0$.

Hence, for $t \leq \frac{1}{2p_1}$, we have

$$\mathbb{E}G(x_t) - F(x^*) \geq \mathbb{E}\left[G(x_t) - G(x^*)\Big|\frac{1}{2p_1} < T\right]\mathbb{P}\left(\frac{1}{2p_1} < T\right)$$
$$= \frac{LB^2}{2}\mathbb{P}\left(\frac{1}{2p_1} < T\right).$$

Note that $T$ follows a geometric distribution with success probability $p_1 \leq 1/n$, and

$$\mathbb{P}\left(T > \frac{1}{2p_1}\right) = \mathbb{P}\left(T > \left\lfloor\frac{1}{2p_1}\right\rfloor\right) = (1-p_1)^{\left\lfloor\frac{1}{2p_1}\right\rfloor}$$
$$\geq (1-p_1)^{\frac{1}{2p_1}} \geq (1-1/n)^{n/2} \geq \frac{1}{2},$$

where the second inequality follows from $h(z) = \frac{\log(1-z)}{2z}$ is a decreasing function.

Thus, for $t \leq \frac{1}{2p_1}$, we have

$$\mathbb{E}G(x_t) - F(x^*) \geq \frac{LB^2}{4} \geq \varepsilon$$

Thus, in order to find $\hat{x} \in \mathbb{R}$ such that $\mathbb{E}F(\hat{x}) - F(x^*) < \varepsilon$, $\mathcal{A}$ needs at least $\frac{1}{2p_1} \geq n/2 = \Omega(n)$ queries to $h_G$.

$\square$

**Proof of Lemma 4.10.** Note that $\{g_i\}_{i=1}^n$ defined in proof of Lemma 4.6 is also $L$-average smooth, so Lemma 4.10 hosts for the same reason. $\square$

**Proof of Lemma 4.3.** Let $B = \sqrt{2\Delta/L}$. Then $\varepsilon/\Delta \leq 1/2$ is equivalent to $\varepsilon \leq LB^2/4$. Note that $\{g_i\}_{i=1}^n$ defined in proof of Lemma 4.6 is also $\mu$-strongly convex for any $\mu \leq L$, and satisfy $|G(0) - G(x^*)| = \Delta$. Therefore Lemma 4.3 hosts for the same reason. $\square$

**Remark H.1.** *Suppose that*

$$\frac{\varepsilon}{\Delta} > \frac{1}{9}\left(\frac{\alpha-1}{\alpha+1}\right)^2, \alpha = \sqrt{2\frac{\kappa-1}{n}+1}.$$

*1. If $\kappa \geq n/2 + 1$, then we have $\alpha \geq \sqrt{2}$ and*

$$\left(n + \sqrt{\kappa n}\right)\log\left(\frac{\Delta}{9\varepsilon}\right) \leq 2\left(n + \sqrt{\kappa n}\right)\log\left(\frac{\alpha+1}{\alpha-1}\right)$$
$$\leq \frac{4\left(n+\sqrt{\kappa n}\right)}{\alpha-1} = \mathcal{O}(n) + \frac{4\sqrt{\kappa n}}{(1-\sqrt{2}/2)\alpha}$$
$$\leq \mathcal{O}(n) + \frac{4}{\sqrt{2}-1}\frac{\sqrt{\kappa n}}{\sqrt{\kappa/n}} = \mathcal{O}(n),$$

*where the second inequality follows from $\log(1+x) \leq x$ and the last inequality is according to $\alpha \geq \sqrt{2\kappa/n}$. That is*

$$\Omega(n) = \Omega\left(\left(n + \sqrt{\kappa n}\right)\log\left(\frac{\Delta}{9\varepsilon}\right)\right).$$

*2. If $2 \leq L/\mu < n/2 + 1$, then we have*

$$\left(\frac{n}{1+\log(n\mu/L)}\right)\log\left(\frac{\Delta}{9\varepsilon}\right) \leq \left(\frac{n}{1+\log(n\mu/L)}\right)\left(2\log\left(\frac{\alpha+1}{\alpha-1}\right)\right)$$
$$\leq \left(\frac{n}{1+\log(n\mu/L)}\right)\left(2\log\left(\frac{(2\sqrt{2}-1)n}{L/\mu}\right)\right) = \mathcal{O}(n),$$

*where the second inequality follows from (22). That is*

$$\Omega(n) = \Omega\left(\left(\frac{n}{1 + \log(n\mu/L)}\right)\log\left(\frac{\Delta}{9\varepsilon}\right) + n\right).$$

## I    DETAILED PROOF FOR SECTION 4.3

***Proof of Proposition 4.7.***

1. It is easily to check that $F_{SC}(\boldsymbol{x})$ is $\mu$-strongly convex. Following from Proposition 2.5, then $\{f_{SC,i}\}_{i=1}^{n}$ is $\hat{L}$-average smooth, where

$$\hat{L} = \sqrt{\frac{16}{n}\left[\left(\frac{L+\mu}{4}\right)^2 + \left(\frac{L-\mu}{4}\right)^2\right] + \mu^2}$$

$$= \sqrt{\frac{2(L^2+\mu^2)}{n} + \mu^2} = L'.$$

2. Clearly, $L = \sqrt{\frac{n(L'^2 - \mu^2)}{2} - \mu^2} \leq \sqrt{\frac{n}{2}}L'$.

   Furthermore, according to $\frac{L'}{\mu} \geq \sqrt{\frac{3}{n}}(\frac{n}{2} + 1)$, we have

$$L^2 - \frac{n}{3}L'^2 = \frac{n}{2}(L'^2 - \mu^2) - \mu^2 - \frac{n}{3}L'^2$$

$$= \frac{1}{2}\left(\frac{n}{2} + 1\right)^2 \mu^2 - \frac{n+2}{2}\mu^2$$

$$= \left(\frac{n^2}{8} - \frac{1}{2}\right)\mu^2 \geq 0,$$

   and, $L/\mu \geq \sqrt{\frac{n}{3}}L'/\mu \geq n/2 + 1$.

$\square$

***Proof of Theorem 4.8.*** By 2nd property of Proposition 4.7, we know that $L/\mu \geq n/2 + 1$. Moreover,

$$m = \frac{1}{4}\left(\sqrt{\sqrt{\frac{2}{n}}\frac{L'}{\mu} + 1}\right)\log\left(\frac{\Delta}{9\varepsilon}\right) + 1$$

$$\geq \frac{1}{4}\left(\sqrt{2\frac{L/\mu - 1}{n} + 1}\right)\log\left(\frac{\Delta}{9\varepsilon}\right) + 1,$$

Then, by Theorem 4.2 [5], in order to find $\hat{\boldsymbol{x}} \in \mathbb{R}^m$ such that $\mathbb{E}F_{SC}(\hat{\boldsymbol{x}}) - F_{SC}(\boldsymbol{x}^*) < \varepsilon$, $\mathcal{A}$ needs at least $N$ queries to $h_{F_{SC}}$, where

$$N = \Omega\left(\left(n + \sqrt{\frac{nL}{\mu}}\right)\log\left(\frac{\Delta}{\varepsilon}\right)\right)$$

$$= \Omega\left(\left(n + \sqrt{\frac{n\sqrt{n/3}L'}{\mu}}\right)\log\left(\frac{\Delta}{\varepsilon}\right)\right)$$

$$= \Omega\left(\left(n + n^{3/4}\sqrt{\frac{L'}{\mu}}\right)\log\left(\frac{\Delta}{\varepsilon}\right)\right).$$

$\square$

---

[5]By the proof of Theorem 4.2, a larger dimension $m$ does not affect the conclusion of the theorem.

***Proof of Theorem 4.9.*** Note that

$$\varepsilon \le \frac{\sqrt{2}}{768} \frac{B^2 L'}{\sqrt{n}} = \frac{B^2 L}{384n},$$

$$m = \left\lfloor \frac{\sqrt[4]{18}}{12} B n^{-1/4} \sqrt{\frac{L'}{\varepsilon}} \right\rfloor - 1 = \left\lfloor \sqrt{\frac{B^2 L}{24n\varepsilon}} \right\rfloor - 1.$$

By Theorem 4.5, in order to find $\hat{\boldsymbol{x}} \in \mathbb{R}^m$ such that $\mathbb{E} F_C(\hat{\boldsymbol{x}}) - F_C(\boldsymbol{x}^*) < \varepsilon$, $\mathcal{A}$ needs at least $N$ queries to $h_{F_C}$, where

$$N = \Omega\left( n + B\sqrt{\frac{nL}{\varepsilon}} \right)$$

$$= \Omega\left( n + B\sqrt{\frac{n\sqrt{n/2}L'}{\varepsilon}} \right)$$

$$= \Omega\left( n + Bn^{3/4}\sqrt{\frac{L'}{\varepsilon}} \right).$$

$\square$

## J  NON-CONVEX CASE

In non-convex case, our goal is to find an $\varepsilon$-approximate stationary point $\hat{\boldsymbol{x}}$ of our objective function $f$, which satisfies

$$\|\nabla f(\hat{\boldsymbol{x}})\|_2 \le \varepsilon. \tag{26}$$

### J.1  PRELIMINARIES

We first introduce a general concept about smoothness.

**Definition J.1.** *For any differentiable function* $f : \mathbb{R}^{m+1} \to \mathbb{R}$, *we say* $f$ *is* $(l, L)$-*smooth, if for any* $\boldsymbol{x}, \boldsymbol{y} \in \mathbb{R}^m$ *we have*

$$\frac{l}{2} \|\boldsymbol{x} - \boldsymbol{y}\|_2^2 \le f(\boldsymbol{x}) - f(\boldsymbol{y}) - \langle \nabla f(\boldsymbol{y}), \boldsymbol{x} - \boldsymbol{y} \rangle \le \frac{L}{2} \|\boldsymbol{x} - \boldsymbol{y}\|_2^2,$$

*where* $L > 0, l \in \mathbb{R}$.

Especially, if $f$ is $L$-smooth, then it can be checked that $f$ is $(-L, L)$-smooth.

If $f$ is $(-\sigma, L)$-smooth, in order to make the operator $\text{prox}_f^\gamma$ valid, we set $\frac{1}{\gamma} > \sigma$ to ensure the function

$$\hat{f}(\boldsymbol{u}) \triangleq f(\boldsymbol{u}) + \frac{1}{2\gamma} \|\boldsymbol{x} - \boldsymbol{u}\|_2^2$$

is a convex function.

Next, we introduce a class of function which is original proposed in (Carmon et al., 2017). Let $G_{\text{NC}} : \mathbb{R}^{m+1} \to \mathbb{R}$ be

$$G_{\text{NC}}(\boldsymbol{x}; \alpha, m) = \frac{1}{2} \left\| \boldsymbol{B}(m + 1, \sqrt[4]{\alpha})\boldsymbol{x} \right\|_2^2 - \sqrt{\alpha}\langle \boldsymbol{e}_1, \boldsymbol{x} \rangle + \alpha \sum_{i=1}^{m} \Gamma(x_i),$$

where the non-convex function $\Gamma : \mathbb{R} \to \mathbb{R}$ is

$$\Gamma(x) \triangleq 120 \int_1^x \frac{t^2(t - 1)}{1 + t^2} dt. \tag{27}$$

We need following properties about $G_{\text{NC}}(\boldsymbol{x}; \alpha, m)$.

**Proposition J.2** (Lemmas 3,4, Carmon et al. (2017)). *For any $0 < \alpha \leq 1$, it holds that*

1. *$\Gamma(x)$ is $(-45(\sqrt{3} - 1), 180)$-smooth and $G_{NC}(\boldsymbol{x}; \alpha, m)$ is $(-45(\sqrt{3} - 1)\alpha, 4 + 180\alpha)$-smooth.*

2. *$G_{NC}(\boldsymbol{0}; \alpha, m) - \min_{\boldsymbol{x} \in \mathbb{R}^{m+1}} G_{NC}(\boldsymbol{x}; \alpha, m) \leq \sqrt{\alpha}/2 + 10\alpha m$.*

3. *For $\boldsymbol{x}$ which satisfies that $x_m = x_{m+1} = 0$, we have*

$$\|\nabla G_{NC}(\boldsymbol{x}; \alpha, m)\|_2 \geq \alpha^{3/4}/4.$$

## J.2   OUR RESULT

**Theorem J.3.** *For any PIFO algorithm $\mathcal{A}$ and any $L, \sigma, n, \Delta, \varepsilon$ such that $\varepsilon^2 \leq \frac{\Delta L \alpha}{81648n}$, there exist a dimension $d = \left\lfloor \frac{\Delta L \sqrt{\alpha}}{40824n\varepsilon^2} \right\rfloor + 1$ and $n$ $(-\sigma, L)$-smooth nonconvex functions $\{f_i : \mathbb{R}^d \to \mathbb{R}\}_{i=1}^n$ such that $f(\boldsymbol{x}_0) - f(\boldsymbol{x}^*) \leq \Delta$. In order to find $\hat{\boldsymbol{x}} \in \mathbb{R}^d$ such that $\mathbb{E} \|\nabla f(\hat{\boldsymbol{x}})\|_2 < \varepsilon$, $\mathcal{A}$ needs at least $\Omega\left(\frac{\Delta L \sqrt{\alpha}}{\varepsilon^2}\right)$ queries to $h_f$, where we set $\alpha = \min\left\{1, \frac{(\sqrt{3}+1)n\sigma}{30L}, \frac{n}{180}\right\}$.*

**Remark J.4.** *For $n > 180$, we have*

$$\Omega\left(\frac{\Delta L \sqrt{\alpha}}{\varepsilon^2}\right) = \Omega\left(\frac{\Delta}{\varepsilon^2} \min\left\{L, \sqrt{\frac{\sqrt{3}+1}{30}} \sqrt{n\sigma L}, \frac{\sqrt{n}L}{\sqrt{180}}\right\}\right) = \Omega\left(\frac{\Delta}{\varepsilon^2} \min\{L, \sqrt{n\sigma L}\}\right).$$

*Thus, our result is comparable to the one of Zhou and Gu's result (their result only related to IFO algorithms, so our result is more strong), but our construction only requires the dimension be $\mathcal{O}\left(1 + \frac{\Delta}{\varepsilon^2} \min\{L/n, \sqrt{\sigma L/n}\}\right)$, which is much smaller than $\mathcal{O}\left(\frac{\Delta}{\varepsilon^2} \min\{L, \sqrt{n\sigma L}\}\right)$ in (Zhou and Gu, 2019).*

## J.3   CONSTRUCTIONS

Consider

$$F(\boldsymbol{x}; \alpha, m, \lambda, \beta) = \lambda G_{NC}(\boldsymbol{x}/\beta; \alpha, m). \tag{28}$$

Similar to our construction we introduced in Section 2, we denote the $l$-th row of the matrix $\boldsymbol{B}(m + 1, \sqrt[4]{\alpha})$ by $\boldsymbol{b}_l$ and

$$\mathcal{L}_i = \{l : 1 \leq l \leq m, m + 1 - l \equiv i(\mathrm{mod}\ n)\}, i = 1, 2, \cdots, n. \tag{29}$$

Let $\mathcal{G}_k = \mathrm{span}\{\boldsymbol{e}_1, \boldsymbol{e}_2, \cdots, \boldsymbol{e}_k\}, 1 \leq k \leq m, \mathcal{G}_0 = \{\boldsymbol{0}\}$ and compose $F(\boldsymbol{x}; \alpha, m, \lambda, \beta)$ to

$$\begin{cases} f_1(\boldsymbol{x}; \alpha, m, \lambda, \beta) = \frac{\lambda n}{2\beta^2} \sum\limits_{l \in \mathcal{L}_i} \left\|\boldsymbol{b}_l^\top \boldsymbol{x}\right\|_2^2 - \frac{\lambda n \sqrt{\alpha}}{\beta} \langle \boldsymbol{e}_1, \boldsymbol{x} \rangle + \lambda \alpha \sum\limits_{i=1}^m \Gamma(x_i/\beta), \\ f_i(\boldsymbol{x}; \alpha, m, \lambda, \beta) = \frac{\lambda n}{2\beta^2} \sum\limits_{l \in \mathcal{L}_i} \left\|\boldsymbol{b}_l^\top \boldsymbol{x}\right\|_2^2 + \lambda \alpha \sum\limits_{i=1}^m \Gamma(x_i/\beta), \text{ for } i \geq 2. \end{cases} \tag{30}$$

Clearly, $F(\boldsymbol{x}; \alpha, m, \lambda, \beta) = \frac{1}{n} \sum_{i=1}^n f_i(\boldsymbol{x}; \alpha, m, \lambda, \beta)$. Moreover, by Proposition J.2, we have following properties about $F(\boldsymbol{x}; \alpha, m, \lambda, \beta)$ and $\{f_i(\boldsymbol{x}; \alpha, m, \lambda, \beta)\}_{i=1}^n$.

**Proposition J.5.** *For any $0 < \alpha \leq 1$, it holds that*

1. *$f_i(\boldsymbol{x}; \alpha, m, \lambda, \beta)$ is $\left(\frac{-45(\sqrt{3}-1)\alpha\lambda}{\beta^2}, \frac{(2n+180\alpha)\lambda}{\beta^2}\right)$-smooth.*

2. *$F(\boldsymbol{0}; \alpha, m, \lambda, \beta) - \min_{\boldsymbol{x} \in \mathbb{R}^{m+1}} F(\boldsymbol{x}; \alpha, m, \lambda, \beta) \leq \lambda(\sqrt{\alpha}/2 + 10\alpha m)$.*

3. *For $\boldsymbol{x}$ which satisfies that $x_m = x_{m+1} = 0$, we have*

$$\|\nabla F(\boldsymbol{x}; \alpha, m, \lambda, \beta)\|_2 \geq \frac{\alpha^{3/4}\lambda}{4\beta}.$$

Similar to Lemma 2.6, similar conclusion hosts for $\{f_i(\boldsymbol{x}; \alpha, m, \lambda, \beta)\}_{i=1}^n$.

**Lemma J.6.** *For $\boldsymbol{x} \in \mathcal{F}_k$, $0 \le k < m$ and $\gamma < \frac{\sqrt{2}+1}{60} \frac{\beta^2}{\lambda \alpha}$, we have*

$$\nabla f_i(\boldsymbol{x}; \alpha, m, \lambda, \beta), \text{prox}_{f_i}^\gamma(\boldsymbol{x}) \in \begin{cases} \mathcal{G}_{k+1}, & \text{if } k \equiv i - 1 (\text{mod } n), \\ \mathcal{G}_k, & \text{otherwise.} \end{cases}$$

*Proof.* Let $G(\boldsymbol{x}) \triangleq \sum\limits_{i=1}^m \Gamma(x_i)$ and $\Gamma'(x)$ be the derivative of $\Gamma(x)$.

First note that $\Gamma'(0) = 0$, so if $\boldsymbol{x} \in \mathcal{G}_k$, then

$$\nabla G(\boldsymbol{x}) = \left(\Gamma'(x_1), \Gamma'(x_2), \cdots, \Gamma'(x_m)\right)^\top \in \mathcal{G}_k.$$

Moreover, for $\boldsymbol{x} \in \mathcal{F}_G$ ($k \ge 1$), we have

$$\boldsymbol{b}_l^\top \boldsymbol{x} = 0 \text{ for } l < m - k,$$
$$\boldsymbol{b}_l \in \mathcal{G}_k \text{ for } l > m - k,$$
$$\boldsymbol{b}_{m-k} \in \mathcal{G}_{k+1}.$$

Consequently, for $l \ne m - k$, $\boldsymbol{b}_l \boldsymbol{b}_l^\top \boldsymbol{x} = (\boldsymbol{b}_l^\top \boldsymbol{x}) \boldsymbol{b}_l \in \mathcal{G}_k$, and $\boldsymbol{b}_{m-k} \boldsymbol{b}_{m-k}^\top \boldsymbol{x} \in \mathcal{G}_{k+1}$.

For $k = 0$, we have $\boldsymbol{x} = \boldsymbol{0}$, and

$$\nabla f_1(\boldsymbol{x}) = \lambda n \sqrt{\alpha}/\beta \; \boldsymbol{e}_1 \in \mathcal{G}_1,$$
$$\nabla f_j(\boldsymbol{x}) = \boldsymbol{0} \; (j \ge 2).$$

For $k \ge 1$, we suppose that $m - k \in \mathcal{L}_i$. Since

$$\nabla f_j(\boldsymbol{x}) = \frac{\lambda n}{\beta^2} \sum_{l \in \mathcal{L}_j} \boldsymbol{b}_l^\top \boldsymbol{b}_l \boldsymbol{x} + \frac{\lambda \alpha}{\beta} \nabla G(\boldsymbol{x}/\beta) - \eta_j \boldsymbol{e}_1,$$

where $\eta_1 = \lambda n \sqrt{\alpha}/\beta$, $\eta_j = 0$ for $j \ge 2$.
Hence, $\nabla f_i(\boldsymbol{x}) \in \mathcal{F}_{k+1}$ and $\nabla f_j(\boldsymbol{x}) \in \mathcal{F}_k$ ($j \ne i$).

Now, we turn to consider $\boldsymbol{v} = \text{prox}_{f_j}^\gamma(\boldsymbol{x})$.

We have

$$\nabla f_j(\boldsymbol{v}) + \frac{1}{\gamma}(\boldsymbol{v} - \boldsymbol{x}) = \boldsymbol{0},$$

that is

$$\left(\frac{\lambda n}{\beta^2} \sum_{l \in \mathcal{L}_j} \boldsymbol{b}_l^\top \boldsymbol{b}_l + \frac{1}{\gamma} \boldsymbol{I}\right) \boldsymbol{v} + \frac{\lambda \alpha}{\beta} \nabla G(\boldsymbol{v}/\beta) = \eta_j \boldsymbol{e}_1 + \frac{1}{\gamma} \boldsymbol{x}. \tag{31}$$

Denote

$$\boldsymbol{A} = \frac{\lambda n}{\beta} \sum_{l \in \mathcal{L}_j} \boldsymbol{b}_l^\top \boldsymbol{b}_l + \frac{\beta}{\gamma} \boldsymbol{I}, \; \boldsymbol{u} = \frac{1}{\beta} \boldsymbol{v}, \; \boldsymbol{y} = \eta_j \boldsymbol{e}_1 + \frac{1}{\gamma} \boldsymbol{x},$$

then we have

$$\boldsymbol{A} \boldsymbol{u} + \frac{\lambda \alpha}{\beta} \nabla G(\boldsymbol{u}) = \boldsymbol{y}. \tag{32}$$

Next, if $s$ satisfies

$$\begin{cases} s > \max\{1, k\} & \text{for } j = 1, \\ s > k & \text{for } j > 1, \end{cases} \tag{33}$$

then we know that the $s$-th element of $\boldsymbol{y}$ is 0.

If $s$ satisfies (33) and $m - s \in \mathcal{L}_j$, then the $s$-th and $(s + 1)$-th elements of $\boldsymbol{Au}$ is $((\xi + \beta/\gamma)u_s - \xi u_{s+1})$ and $(-\xi u_s + (\xi + \beta/\gamma)u_{s+1})$ respectively where $\xi = \lambda n/\beta$. So by Equation (32), we have

$$
\begin{cases}
\frac{\beta}{\gamma}u_s + \xi(u_s - u_{s+1}) + \frac{120\lambda\alpha}{\beta}\frac{u_s^2(u_s-1)}{1+u_s^2} = 0. \\
\frac{\beta}{\gamma}u_{s+1} + \xi(u_{s+1} - u_s) + \frac{120\lambda\alpha}{\beta}\frac{u_{s+1}^2(u_{s+1}-1)}{1+u_{s+1}^2} = 0.
\end{cases}
$$

Following from Lemma J.9, for $\frac{120\lambda\alpha}{\beta} < \frac{(2+2\sqrt{2})\beta}{\gamma}$, we have $u_s = u_{s+1} = 0$.
That is

1. if $m - s \in \mathcal{L}_j$ and $s$ satisfies (33), then $u_s = 0$.

2. if $m - s + 1 \in \mathcal{L}_j$ and $s - 1$ satisfies (33), then $u_s = 0$.

For $s$ which satisfies (33), if $m - s \notin \mathcal{L}_j$ and $m - s + 1 \notin \mathcal{L}_j$, then the $s$-th element of $\boldsymbol{Au}$ is $(\beta/\gamma\, u_s)$. Similarly, by Equation (32), we have

$$
\frac{\beta}{\gamma}u_s + \frac{120\lambda\alpha}{\beta}\frac{u_s^2(u_s - 1)}{1 + u_s^2} = 0.
$$

Following from Lemma J.8, for $\frac{120\lambda\alpha}{\beta} < \frac{(2+2\sqrt{2})\beta}{\gamma}$, we have $u_s = 0$.

Therefore, we can conclude that

1. if $s - 1$ satisfies (33), then $u_s = 0$.

2. if $s$ satisfies (33) and $m - s + 1 \notin \mathcal{L}_j$, then $u_s = 0$.

Moreover, we have that

1. if $k = 0$ and $j = 1$, then $m - 1, m - 2 \notin \mathcal{L}_j$, so $u_2 = 0$.

2. if $k = 0$ and $j > 1$, then for $s = 1$, we have $m - s + 1 \notin \mathcal{L}_j$, so $u_1 = 0$.

3. if $k = 0$, then for $s > 2$, we have $s - 1 > 1$ satisfies (33), so $u_s = 0$.

4. if $k > 0$, then for $s > k + 1$, we have $s - 1 > k$ satisfies (33), so $u_s = 0$.

5. if $k > 0$ and $m - k \notin \mathcal{L}_j$, then for $s = k + 1$, we have $m - s + 1 \notin \mathcal{L}_j$, so $u_{k+1} = 0$.

In short,

1. if $k = 0$ and $j > 1$, then $\boldsymbol{u} \in \mathcal{G}_0$.

2. if $k = 0$ and $j = 1$, then $\boldsymbol{u} \in \mathcal{G}_1$.

3. if $k > 1$ and $m - k \notin \mathcal{L}_j$, then $\boldsymbol{u} \in \mathcal{G}_k$.

4. if $k > 1$ and $m - k \in \mathcal{L}_j$, then $\boldsymbol{u} \in \mathcal{G}_{k+1}$.

$\square$

**Remark J.7.** *In order to make the operator* $\mathrm{prox}_{f_i}^\gamma$ *valid,* $\gamma$ *need to satisfy*

$$
\gamma < \frac{\sqrt{3}+1}{90}\frac{\beta^2}{\lambda\alpha} < \frac{\sqrt{2}+1}{60}\frac{\beta^2}{\lambda\alpha}.
$$

*So for any valid PIFO call, the condition about* $\gamma$ *in Lemma J.6 must be satisfied.*

**Lemma J.8.** *Suppose that $0 < \lambda_2 < (2 + 2\sqrt{2})\lambda_1$, then $z = 0$ is the only real solution to the equation*

$$\lambda_1 z + \lambda_2 \frac{z^2(z-1)}{1+z^2} = 0. \tag{34}$$

*Proof.* Since $0 < \lambda_2 < (2 + 2\sqrt{2})\lambda_1$, we have

$$\lambda_2^2 - 4\lambda_1(\lambda_1 + \lambda_2) < 0,$$

and consequently, for any $z$, $(\lambda_1 + \lambda_2)z^2 - \lambda_2 z + \lambda_1 > 0$.

On the other hand, we can rewrite Equation (34) as

$$z\big((\lambda_1 + \lambda_2)z^2 - \lambda_2 z + \lambda_1\big) = 0.$$

Clearly, $z = 0$ is the only real solution to Equation (34).

$\square$

**Lemma J.9.** *Suppose that $0 < \lambda_2 < (2 + 2\sqrt{2})\lambda_1$ and $\lambda_3 > 0$, then $z_1 = z_2 = 0$ is the only real solution to the equation*

$$\begin{cases} \lambda_1 z_1 + \lambda_3(z_1 - z_2) + \lambda_2 \frac{z_1^2(z_1-1)}{1+z_1^2} = 0. \\ \lambda_1 z_2 + \lambda_3(z_2 - z_1) + \lambda_2 \frac{z_2^2(z_2-1)}{1+z_2^2} = 0. \end{cases} \tag{35}$$

*Proof.* If $z_1 = 0$, then $z_2 = 0$. So let assume that $z_1 z_2 \neq 0$. Rewrite the first equation of Equation (35) as

$$\frac{\lambda_1 + \lambda_3}{\lambda_3} + \frac{\lambda_2}{\lambda_3}\frac{z_1(z_1-1)}{1+z_1^2} = \frac{z_2}{z_1}$$

Note that

$$\frac{1-\sqrt{2}}{2} \leq \frac{z(z-1)}{1+z^2}.$$

Thus, we have

$$\frac{\lambda_1 + \lambda_3}{\lambda_3} + \frac{\lambda_2}{\lambda_3}\frac{1-\sqrt{2}}{2} \leq \frac{z_2}{z_1}.$$

Similarly, it also holds

$$\frac{\lambda_1 + \lambda_3}{\lambda_3} + \frac{\lambda_2}{\lambda_3}\frac{1-\sqrt{2}}{2} \leq \frac{z_1}{z_2}.$$

By $0 < \lambda_2 < (2 + 2\sqrt{2})\lambda_1$, we know that $\lambda_1 + \frac{1-\sqrt{2}}{2}\lambda_2 > 0$. Thus

$$\frac{\lambda_1 + \lambda_3}{\lambda_3} + \frac{\lambda_2}{\lambda_3}\frac{1-\sqrt{2}}{2} > 1.$$

Since $z_1/z_2 > 1$ and $z_2/z_1 > 1$ can not hold at the same time, so we get a contradiction. $\square$

Following from Lemma J.6, we know following Lemma which is similar to Lemma 2.9.

**Lemma J.10.** *If $M \geq 1$ satisfies $\min_{x \in \mathcal{G}_M} \|\nabla F(x)\|_2 \geq 9\varepsilon$ and $N = n(M+1)/4$, then we have*

$$\min_{t \leq N} \mathbb{E}\|\nabla F(x_t)\|_2 \geq \varepsilon.$$

**Theorem J.11.** *Set*

$$\alpha = \min\left\{1, \frac{(\sqrt{3}+1)n\sigma}{30L}, \frac{n}{180}\right\},$$

$$\lambda = \frac{3888n\varepsilon^2}{L\alpha^{3/2}},$$

$$\beta = \sqrt{3\lambda n/L},$$

$$m = \left\lfloor \frac{\Delta L\sqrt{\alpha}}{40824n\varepsilon^2} \right\rfloor$$

*Suppose that* $\varepsilon^2 \leq \frac{\Delta L\alpha}{81648n}$*. In order to find* $\hat{\boldsymbol{x}} \in \mathbb{R}^{m+1}$ *such that* $\mathbb{E}\left\|\nabla F(\hat{\boldsymbol{x}})\right\|_2 < \varepsilon$*, PIFO algorithm* $\mathcal{A}$ *needs at least* $\Omega\left(\frac{\Delta L\sqrt{\alpha}}{\varepsilon^2}\right)$ *queries to* $h_F$*.*

*Proof.* First, note that $f_i$ is $(-l_1, l_2)$-smooth, where

$$l_1 = \frac{45(\sqrt{3}-1)\alpha\lambda}{\beta^2} = \frac{45(\sqrt{3}-1)L}{3n}\alpha \leq \frac{45(\sqrt{3}-1)L}{3n}\frac{(\sqrt{3}+1)n\sigma}{30L} = \sigma,$$

$$l_2 = \frac{(2n+180\alpha)\lambda}{\beta^2} = \frac{L}{3n}(2n+180\alpha) \leq L.$$

Thus each $f_i$ is $(-\sigma, L)$-smooth.

Next, observe that

$$F(\boldsymbol{x}_0) - F(\boldsymbol{x}^*) \leq \lambda(\sqrt{\alpha}/2 + 10\alpha m) = \frac{1944n\varepsilon^2}{L\alpha} + \frac{38880n\varepsilon^2}{L\sqrt{\alpha}}m$$

$$\leq \frac{1944}{40824}\Delta + \frac{38880}{40824}\Delta = \Delta.$$

For $M = m - 1$, we know that

$$\min_{\boldsymbol{x} \in \mathcal{G}_M} \left\|\nabla F(\boldsymbol{x})\right\|_2 \geq \frac{\alpha^{3/4}\lambda}{4\beta} = \frac{\alpha^{3/4}\lambda}{4\sqrt{3\lambda n/L}} = \sqrt{\frac{\lambda L}{3n}}\frac{\alpha^{3/4}}{4} = 9\varepsilon.$$

With recalling Lemma J.10, in order to find $\hat{\boldsymbol{x}} \in \mathbb{R}^{m+1}$ such that $\mathbb{E}\left\|\nabla F(\hat{\boldsymbol{x}})\right\|_2 < \varepsilon$, PIFO algorithm $\mathcal{A}$ needs at least $N$ queries to $h_F$, where

$$N = n(M+1)/4 = nm/4 = \Omega\left(\frac{\Delta L\sqrt{\alpha}}{\varepsilon^2}\right).$$

At last, we need to ensure that $m \geq 2$. By $\varepsilon^2 \leq \frac{\Delta L\alpha}{81648n}$, we have

$$\frac{\Delta L\sqrt{\alpha}}{40824n\varepsilon^2} \geq \frac{\Delta L\alpha}{40824n\varepsilon^2} \geq 2,$$

and consequently $m \geq 2$. $\square$

