# OpenReview forum: "A Novel Analysis Framework of Lower Complexity Bounds for Finite-Sum Optimization"
_ICLR.cc/2020/Conference — Reject_

### Official Review · AnonReviewer1 · 2019-10-22
**Official Blind Review #1**

**Rating:** 3

**Review:**

The authors prove lower bounds on the number of queries required for optimizing sums of convex functions.  They consider more powerful queries than the usual queries that provide function evaluation/gradient pairs for chosen summands.  As was done in [1] (which is cited in the submission), in this work algorithms can also get the answer to a
"prox query" solving a regularized optimization problem for the chosen summand at the chosen point.  For different classes of functions obtained through smoothness and (strong) convexity constraints, the lower bounds are on
the number of queries needed by an algorithm to guarantee to approximate the minimum.

The main result is for the case that the summands are mu-strongly convex and L-smooth.  Bounds for this case are often
given in terms of kappa = L/mu.  An upper bound of O( (n + sqrt(kappa n) ) log(1/eps)) is known, and
[1] had proved a lower bound of Omega( n + sqrt(kappa n)  log(1/eps)), which matches the second term of the upper bound, but leaves a log-factor gap for the first.  This paper proves an Omega( (n + sqrt(kappa n) ) log(1/eps))  lower bound, but for a restricted class of algorithms that fix a probability distribution over the summands ahead of time, and randomize by repeatedly sampling independently at random from this fixed distribution.  The iterates of the algorithm are also constrained to be in the span of the answers to previous queries.  Thus, this new result is incomparable in strength with the result in [1].  Also, the authors of this paper mention early in the paper that kappa is often large relative to n.
But even if kappa is on the same order as n, the second term of the upper bound dominates the first, and is matched by the lower bound in [1].

The authors point to some new techniques in their analysis.  I can see some new elements, but my knowledge of the previous work in this area is not deep enough to evaluate technical novelty very well.

I have some question about the extent to which this work is in scope for ICLR.  An argument could go that since stochastic gradient methods are so important to deep learning, study of the foundations and limitations of those methods is in scope.
But a lower bound for the convex case seems to be stretching this a little far.

This seems like a somewhat incremental contribution that would be of interest to a smallish subset of ICLR attendees.

[1] Blake Woodworth and Nathan Srebro. Tight complexity bounds for
optimizing composite objectives. In NIPS, 2016.

**Experience Assessment:**

I have read many papers in this area.

**Review Assessment: Checking Correctness Of Derivations And Theory:**

I assessed the sensibility of the derivations and theory.

**Review Assessment: Checking Correctness Of Experiments:**

N/A

**Review Assessment: Thoroughness In Paper Reading:**

I read the paper at least twice and used my best judgement in assessing the paper.

---

> ### Author Response · Authors · 2019-11-15
> **Thanks for your comments!**
>
> Thanks for the reviewer's insightful and helpful comments.
>
> 1.  We have appended a new lower bound in the case of $\kappa = \mathcal{O}(n)$, which matches the upper bound of IFO algorithm (Hannah et al., 2018)  (see Table 1, Theorem 3.1 and Section 4.1 in our latest version of the paper). Please note that the our lower bound analysis includes PIFO algorithms, while previous results (Hannah et al., 2018) only consider IFO one in this case.
>
> 2. We must emphasize that the constructions in our proof are novel and different from previous work, which makes our results  stronger. We provide the detailed comparison of our technique  with existing proofs in Appendix B of our latest version.
>
> Briefly speaking
> a) The analysis in (Lan and Zhou, 2017; Zhou and Gu, 2019) employed an aggregation framework while this paper proposed a new decomposition framework. The aggregation one is only valid for analyzing IFO algorithms, while our decomposition framework can be easily extended to show the lower bound of PIFO algorithms.
>
> b) The analysis in (Woodworth and Srebro, 2016; Fang et al., 2018) considers a very complicated approach to dealing with the proximal operator (completely different from how to deal with gradient operator). In contrast, our construction holds ''only one'' property (Lemma 2.6) both for proximal and gradient operator, which makes the proof more concise. We also use our technique to prove the tight lower bound of PIFO algorithm when $\kappa = \mathcal{O}(n)$, which is a new result.
>
> 3. It is worth noting that the proposed analysis framework is general and NOT limited to convex optimization, which also can be used to study NON-CONVEX problems. And we have provided the results and proofs about nonconvex optimization in Appendix J.

---

### Official Review · AnonReviewer3 · 2019-10-23
**Official Blind Review #3**

**Rating:** 8

**Review:**

** Summary
The paper derives a novel lower bound on the complexity of optimizing finite-sum convex functions (under different assumptions) using algorithms that have access to point-wise evaluation of the function, its gradient, and proximal information.

** Overall evaluation
Finite-sum convex functions are very common in machine learning problems and how the optimization complexity scales with their properties (e.g., condition number) and the number of components (e.g., number of samples in typical ML problems) is a very important question. This paper addresses the question from a lower bound point of view, showing that there is no proximal incremental first-order algorithm that can optimize such functions at an accuracy level of epsilon in less than a term which depends linearly with number of components n and sqrt(k) (k being the condition number). The paper fills an existing gap in the literature and it achieves two very interesting results:
1- The lower bound now matches an existing upper bound for Point-SAGA, showing that no better algorithm can exist (at least in a worst-case sense).
2- This result also illustrate that proximal algorithms are not necessarily more powerful than first-order methods that only access the gradient of the function. This is also very interesting, as it was still an open question whether proximal information could possibly give an advantage.

The paper is also well written, although some elements could be improved:
1- Def 2.4: the authors consider algorithms where the sampling distribution cannot adapt through iterations. Although this is standard, I am wondering whether adaptivity may buy anything in the performance or whether the lower bound applies to adaptive algorithms as well.
2- Although similar constructions to create worst-case functions were used before in deriving complexity lower bounds, it would be useful to have an intuition about the specific choice made in eg Eq.5/6 and how this enables the refined analysis presented in the paper.
3- More in general, I encourage the authors to illustrate how their techniques compare and differ from previous lower bound proofs.
4- In all theorems, the analysis is done by linking the dimension d to all other parameters of the problem. As pointed out by the authors, the requirements on the dimensionality in the theorems of this paper are milder than previous results. It would be helpful to illustrate how the lower bound would behave when the dimensionality changes and provide an intuition about the specific choice in the theorems

**Experience Assessment:**

I have read many papers in this area.

**Review Assessment: Checking Correctness Of Derivations And Theory:**

I assessed the sensibility of the derivations and theory.

**Review Assessment: Checking Correctness Of Experiments:**

N/A

**Review Assessment: Thoroughness In Paper Reading:**

I read the paper at least twice and used my best judgement in assessing the paper.

---

> ### Author Response · Authors · 2019-11-15
> **Thanks for your comments!**
>
> Thanks for the reviewer's insightful and helpful comments.
>
> 1. We have appended a new lower bound in the case of $\kappa = \mathcal{O}(n)$, which matches the upper bound of IFO algorithm (Hannah et al., 2018)  (see Table 1, Theorem 3.1 and Section 4.1 in our latest version of the paper). Please note that the our lower bound analysis includes PIFO algorithms, while previous results (Hannah et al., 2018) only consider IFO one in this case.
>
> 2. It is interesting to use our framework to analyze the lower bound of algorithms with adaptive sampling. This extension looks not easy and we would like to study it in future work.
>
> 3. The main reason of using the construction Eq.5/6 is the decomposition $r({\bf x})=\sum_{i=1}^n r_i({\bf x})$ (omit the constants) is friendly to the analysis of PIFO algorithms. Concretely, our construction holds ''only one'' property (Lemma 2.6) both for proximal and gradient operator, while the construction of (Lan and Zhou, 2017; Zhou and Gu, 2019) only holds this property for IFO, which leads their construction is invalid to analyze PIFO algorithms.
>
> 4. We have included a comparison with other constructions in Appendix B.
>
> Briefly speaking
> a) The analysis in (Lan and Zhou, 2017; Zhou and Gu, 2019) employed an aggregation framework while this paper proposed a new decomposition framework.
> As we stated in Reply 2, the construction in (Lan and Zhou, 2017; Zhou and Gu, 2019) is only valid for analyzing IFO algorithms, while our decomposition framework can be easily extended to show the lower bound of PIFO algorithms.
>
> b) The analysis in (Woodworth and Srebro, 2016; Fang et al., 2018) considers a very complicated approach to dealing with the proximal operator (completely different from how to deal with gradient operator). In contrast, our construction holds ''only one'' property (Lemma 2.6) both for proximal and gradient operator, which makes the proof more concise. We also use our technique to prove the tight lower bound of PIFO algorithm when $\kappa = \mathcal{O}(n)$, which is a new result.
>
> 5. The construction $f$ of (Lan and Zhou, 2017; Zhou and Gu, 2019) is from ${\mathbb R}^{mn}$ to ${\mathbb R}$ while our $r$ is from ${\mathbb R}^{m}$ to ${\mathbb R}$ (please see the detailed definitions of $r$ and $f$ in Appendix B), which provides an intuitive understanding why our construction requires a smaller dimension.

---

### Official Review · AnonReviewer2 · 2019-10-27
**Official Blind Review #2**

**Rating:** 6

**Review:**

This paper proves a better complexity lower bound for stochastic PIFO optimizers on the problem of finite-sum minimization. The paper assumes that the objective function is the sum of n individual loss functions. It further assumes that (1) the optimizer initializes at a fixed point, and (2) at each iteration, it randomly and independently selects one loss function to update the parameter vector.

To prove the desired lower bound, the paper constructed a group of special loss functions, such that each individual loss depends on only 2 coordinates of the parameter vector (except for the regularization term). By this construction, if the parameter vector is initialized at 0, then the number of non-zero coordinates of it will grow slowly enough so that the parameter vector will stay in some low-dimensional subspace unless a large number of iterations is performed. Using this construction, the authors prove the lower bound for 4 different configurations of optimization problems.

Overall, I think the results are very interesting. Similar ideas (the diagonal matrix used in this paper) have been widely adopted in proving complexity lower bound. The novelty of this paper appears to be that the diagonal matrix is partitioned into n groups to define the individual loss functions. Despite the tight lower bound, the assumption (1) and (2) above seems to be restrictive, but they are necessary for the analysis of this paper. If we allow the optimizer to initialize at a random point, or if the optimizer can adaptively choose the loss function at each iteration based on the parameter trajectory, then the analysis framework no longer applies. This is probably the main limitation of this work.


**Experience Assessment:**

I have read many papers in this area.

**Review Assessment: Checking Correctness Of Derivations And Theory:**

I assessed the sensibility of the derivations and theory.

**Review Assessment: Checking Correctness Of Experiments:**

I did not assess the experiments.

**Review Assessment: Thoroughness In Paper Reading:**

I read the paper at least twice and used my best judgement in assessing the paper.

---

> ### Author Response · Authors · 2019-11-15
> **Thanks for your comments!**
>
> Thanks for the reviewer's insightful and helpful comments.
>
> 1.  We have appended a new lower bound in the case of $\kappa = \mathcal{O}(n)$, which matches the upper bound of IFO algorithm (Hannah et al., 2018)  (see Table 1, Theorem 3.1 and Section 4.1 in our latest version of the paper). Please note that the our lower bound analysis includes PIFO algorithms, while previous results (Hannah et al., 2018) only consider IFO one in this case.
>
> 2. Our framework is valid for any initial point. As we stated at the bottom of Page 3, we can take $\{{\hat f}_i({\bf x}) = f_i({\bf x} + {\bf x}_0)\}_{i=1}^n$ into consideration if the initial point ${\bf x}_0\neq 0$. Then analyzing ${\hat f}_i({\bf x}) $ is similar to analyzing $f_i({\bf x})$ with ${\bf x}_0=0$.
>
> 3. It is interesting to use our framework to analyze the lower bound of algorithms with adaptive sampling. This extension looks not easy and we would like to study it in future work.

---

### Public Comment · ~Sebastian_U_Stich1 · 2019-10-15
**Lower bounds by Hannah et al.**

Interesting paper!
How do your results (and assumptions) compare to e.g. the lower bounds in https://arxiv.org/abs/1805.07786?
Thanks for clarification,

---

> ### Author Response · Authors · 2019-10-18
> **Thanks for reviewing!**
>
> 1. Many thanks for your reviewing.
> 2. The algorithms in Theorem 2 of (Hannah et al., 2018) use IFO while algorithms in our results use PIFO. PIFO provides more information than IFO and it would be potentially more powerful than IFO in first order optimization algorithms. Moreover, we develop a novel analysis framework to deal with PIFO algorithms and our framework is much simpler and more straightforward than the approach in  (Woodworth and Srebro, 2016) .
> 3. Indeed, (Hannah et al., 2018) obtained a slightly better result for IFO algorithms. However, a subtle adjustment like the approach in (Hannah et al., 2018) also can improve our result to match the upper bound in (Hannah et al., 2018) as well. And we will update the modified results as soon as possible.

---

### Author Response · Authors · 2019-11-15
**General Comments on Latest Revision**

We would like to thanks all official reviewers and Sebastian U Stich for their insightful and helpful comments.
We have appended a new lower bound in the case of $\kappa = \mathcal{O}(n)$ for strongly-convex optimization, which matches the upper bound of IFO algorithm (Hannah et al., 2018)  (see Table 1, Theorem 3.1 and Section 4.1 in our latest version of the paper).
We also have included a comparison with other constructions in Appendix B.

---

### Decision · Program_Chairs · 2019-12-19

**Decision:**

Reject

**Comment:**

The paper considers a lower bound complexity for the convex problems. The reviewers worry about whether the scope of this paper fit in ICLR, the initialization issues, and the novelty and some other problems.